# A Theoretical Understanding of Gradient Bias in Meta-Reinforcement Learning

**Bo Liu**[*]
Institute of Automation,
Chinese Academy of Sciences
benjaminliu.eecs@gmail.com

**Xidong Feng**[*]
University College London
xidong.feng.20@ucl.ac.uk

**Jie Ren**
University of Edinburgh
jieren9806@gmail.com

**Luo Mai**
University of Edinburgh
luo.mai@ed.ac.uk

**Rui Zhu**
DeepMind
ruizhu@google.com

**Haifeng Zhang**
Institute of Automation, CAS
Nanjing Artificial Intelligence Research of IA
haifeng.zhang@ia.ac.cn

**Jun Wang**
University College London
jun.wang@cs.ucl.ac.uk

**Yaodong Yang**[†]
Institute for AI, Peking University
Beijing Institute for General AI
yaodong.yang@pku.edu.cn

## Abstract

Gradient-based Meta-RL (GMRL) refers to methods that maintain two-level optimisation procedures wherein the outer-loop meta-learner guides the inner-loop gradient-based reinforcement learner to achieve fast adaptations. In this paper, we develop a unified framework that describes variations of GMRL algorithms and points out that existing stochastic meta-gradient estimators adopted by GMRL are actually **biased**. Such meta-gradient bias comes from two sources: 1) the compositional bias incurred by the two-level problem structure, which has an upper bound of $O\big(K\alpha^K\hat{\sigma}_{\text{In}}|\tau|^{-0.5}\big)$ *w.r.t.* inner-loop update step $K$, learning rate $\alpha$, estimate variance $\hat{\sigma}_{\text{In}}^2$ and sample size $|\tau|$, and 2) the multi-step Hessian estimation bias $\hat{\Delta}_H$ due to the use of autodiff, which has a polynomial impact $O\big((K-1)(\hat{\Delta}_H)^{K-1}\big)$ on the meta-gradient bias. We study tabular MDPs empirically and offer quantitative evidence that testifies our theoretical findings on existing stochastic meta-gradient estimators. Furthermore, we conduct experiments on Iterated Prisoner's Dilemma and Atari games to show how other methods such as off-policy learning and low-bias estimator can help fix the gradient bias for GMRL algorithms in general.

## 1 Introduction

Meta Learning, also known as learning to learn, is proposed to equip intelligent agents with meta knowledge for fast adaptations [30]. Meta-Reinforcement Learning (RL) algorithms aim to train RL

---

[*]Equal contribution, the order is determined by flipping a coin. See Appendix J for more details.
[†]Corresponding author.

36th Conference on Neural Information Processing Systems (NeurIPS 2022).

agents that can adapt to new tasks with only few examples [5, 10, 16]. For example, MAML-RL [10], a typical Meta-RL algorithm, learns the initial parameters of an agent's policy so that the agent can rapidly adapt to new environments with a limited number of policy-gradient updates. Recently, Meta-RL algorithms have been further developed beyond the scope of learning fast adaptations. An important direction is to conduct online meta-gradient learning for adaptively tuning algorithmic hyper-parameters [39, 41] or designing intrinsic reward [42] during training in one single task. Besides, there are Meta-RL developments that manage to discover new RL algorithms by learning algorithmic components or other fundamental concepts in RL, such as the policy gradient objective [26] or the TD-target [40], which can generalise to solve a distribution of different tasks.

In general, gradient-based Meta-RL (GMRL) tasks can be formulated by a two-level optimisation procedure. This procedure optimises the parameters of an outer-loop meta-learner, whose objective is dependent on a $K$-step policy update process (e.g., stochastic gradient descent) conducted by an inner-loop learner. Formally, this procedure can be written as:

$$
\begin{aligned}
\max_{\boldsymbol{\phi}} J^K(\boldsymbol{\phi}) &:= J^{\text{Out}}(\boldsymbol{\phi}, \boldsymbol{\theta}^K), \\
\text{s.t. } \boldsymbol{\theta}^{i+1} &= \boldsymbol{\theta}^i + \alpha \nabla_{\boldsymbol{\theta}^i} J^{\text{In}}(\boldsymbol{\phi}, \boldsymbol{\theta}^i), i \in \{0, 1 \dots K-1\},
\end{aligned}
\tag{1}
$$

where $\boldsymbol{\theta}$ are inner-loop policy parameters, $\boldsymbol{\phi}$ are meta parameters, $\alpha$ is the learning rate, $J^{\text{In}}$ and $J^{\text{Out}}$ are value functions for the inner and the outer-loop learner.

In solving Eq. (1), estimating the meta-gradient of $\nabla_{\boldsymbol{\phi}} J^K(\boldsymbol{\phi})$ from a two-level optimisation process is non-trivial; how to conduct *accurate* estimation on $\nabla_{\boldsymbol{\phi}} J^K(\boldsymbol{\phi})$ has been a critical yet challenging research problem [23, 29, 33].

In this paper, we point out that the meta-gradient estimators adopted by many recent GMRL methods [26, 40] are in fact biased. We conclude that such bias comes from two sources: (1) the **compositional bias** and (2) the **multi-step Hessian bias**. The compositional bias origins from the discrepancy between the sampled policy gradient $\nabla_{\boldsymbol{\theta}} \hat{J}^{\text{In}}$ and expected policy gradient $\nabla_{\boldsymbol{\theta}} J^{\text{In}}$, and the multi-step Hessian estimation bias occurs due to the biased Hessian estimation $\nabla_{\boldsymbol{\theta}}^2 \hat{J}^{\text{In}}$ resulting from the employment of automatic differentiation in modern GMRL implementations.

There are very few prior work that investigates the bias in GMRL, most of them limited to the Hessian estimation bias in the MAML-RL setting [29, 33]. Our paper investigates the above two biases in a broader setting and applies on generic meta-gradient estimators. For the compositional bias term, we offer the first theoretical analysis on its quantity, based on which we then investigate current mitigation solutions. For the multi-step Hessian bias, we provide rigorous analysis in GMRL settings particularly for those GMRL tasks where complex inner-loop optimisations are involved.

For the rest of the paper, we first introduce a *unified* Meta-RL framework that can describe variations of existing GMRL algorithms. Building on this framework, we offer two theoretical results that **1)** the compositional bias has an upper bound of $O(K\alpha^K \hat{\sigma}_{\text{In}} |\tau|^{-0.5})$ with respect to the inner-loop update step $K$, the learning rate $\alpha$, the estimate variance $\hat{\sigma}_{\text{In}}^2$ and the sample size $|\tau|$, and **2)** the multi-step Hessian bias $\hat{\Delta}_H$ has a polynomial impact of $O((K-1)(\hat{\Delta}_H)^{K-1})$. To validate our theoretical insights on these two biases, we conduct a comprehensive list ablation studies. Experiments over tabular MDP with MAML-RL [10] and LIRPG [42] demonstrate how quantitatively these two biases influence the estimation accuracy, which consolidates our theories.

Furthermore, our theoretical results help understand to what extent existing methods can mitigate theses two bias empirically. For the compositional bias, we show that off-policy learning methods can reduce the inner-loop policy gradient variance and the resulting compositional bias. For the multi-step Hessian bias, we study how the low-variance curvature technique based on Rothfuss et al. [29] can help correct the Hessian bias for general GMRL problems. We test these solutions on environments including iterated Prisoner's Dilemma with off-policy corrected LOLA-DiCE [13], and eight Atari games based on MGRL [39] with the multi-step Hessian correction technique. Experimental results confirm that those bias-correction methods can substantially decrease the meta-gradients bias and improve the overall performance on rewards.

## 2 Related Work

A pioneering work that studies meta-gradient estimation is Al-Shedivat et al. [1] who discussed the biased estimation problem of MAML and proposed the E-MAML sample based formulation to fix the meta-gradient bias. Following work includes Rothfuss et al. [29], Liu et al. [23], Tang et al. [33] that tried to fix the meta-gradient estimation error by reducing the estimation variance so as to improve the performance. Moreover, Foerster et al. [12], Farquhar et al. [8], Mao et al. [24] discussed the higher-order gradient estimation in RL. Recently, Bonnet et al. [4], Vuorio et al. [36] proposed algorithms to balance the bias-variance trade-off for meta-gradient estimates. Within theoretical context of GMRL, most theoretical analysis focuses on convergence to stationary points. Fallah et al. [6] established convergence guarantees of gradient-based meta-learning algorithms for supervised learning with one-step inner-loop update. Ji et al. [18] extended the analysis to multi-step inner loop updates. Fallah et al. [7] established convergence for the E-MAML formulation. Our work is different from all the above prior work from three folds: (1) we study the additional bias term (the compositional bias) (2) we consider different formulation (expected update while sample-based update in [36], refer to Appendix B for more discussions) (3) we study a broader scope of applications that include different GMRL algorithm instantiations and settings.

In the following part, we review existing GMRL algorithms and categorise them into four research topics. We offer one typical example for each topic in Table 1 and further discussed their relationship and how they can be unified into one framework in Sec. 3. We also provide a more self-contained explanation for each topic in Appendix A for readers that are not familiar with GMRL.

**Few-shot RL.** The idea of few-shot RL is to enable RL agent with fast learning ability. Specifically, the RL agent is only allowed to interact with the environment for a few trajectory to conduct task-specific adaptation. Conducting few-shot RL has two approaches: gradient based and context based. Context based Meta-RL involves works [5, 15, 27, 37], which uses neural network to embed the information from few-shot interactions so as to obtain task-relevant context. Gradient based few-shot RL [1, 23, 29] focus on meta-learning the model's initial parameters through meta-gradient descent.

**Opponent Shaping.** Foerster et al. [13], Letcher et al. [22], Kim et al. [19] explicitly models the learning process of opponents in multi-agent learning problems, which can be thought of as GMRL problems since meta-gradient estimation in these works involve differentiation over opponent's policy updates. By modelling opponents' learning process, the multi-agent learning process can reach better social welfare [13, 22] or the ego agent can adapt to a new peer agent [19].

**Single-lifetime Meta-gradient RL.** This line of research focuses on learning online adaptation over algorithmic hyper-parameters to enhance the performance of an RL agent in one single task, such as discount factor [39, 41], intrinsic reward generator [42], auxiliary loss [34], reward shaping mechanism [17], and value correction [44]. The main feature of online meta-gradient RL is that they are under single-lifetime setting [40], meaning that the algorithm only iterates through the whole RL learning procedure in one task rather a distribution of tasks.

**Multi-lifetime Meta-gradient RL.** Compared to the previous single-lifetime settings, "multi-lifetime" refers to settings where agents learn to adapt on a distribution of tasks or environments. The meta-learning target includes policy gradient or TD learning objectives [3, 20, 26], intrinsic reward [43], target value function [40], options in hierarchical RL [35], and recently the design of curriculum in multi-agent learning [9].

## 3 A Unified Framework for Meta-gradient Estimation

In this section, we derive a general formulation for meta-gradient estimation in GMRL. This formulation enables us to conduct general analysis about meta-gradient estimation and we will show how algorithms in the four research topics mentioned in Sec. 2 can be described through it.

We propose that a general GMRL objective with $K$-step inner-loop policy gradient update can be written as the following objective

$$\max_{\boldsymbol{\phi}} J^K(\boldsymbol{\phi}) := J^{\text{Out}}(\boldsymbol{\phi}, \theta^K), \theta^K = \theta^0 + \alpha \sum_{i=0}^{K-1} \nabla_{\theta^i} J^{\text{In}}(\boldsymbol{\phi}, \theta^i). \tag{2}$$

Table 1: Four typical gradient-based Meta-RL (GMRL) algorithms.

| Category | Algorithms | Meta parameter $\phi$ | Inner parameter $\theta$ |
|---|---|---|---|
| Few-shot RL | MAML [10] | Initial Parameter | Initial Parameter |
| Opponent Shaping | LOLA [13] | Ego-agent Policy | Other-agent Policy |
| Single-lifetime MGRL | MGRL [39] | Discount Factor | RL Agent Policy |
| Multi-lifetime MGRL | LPG [26] | LSTM Network | RL Agent Policy |

We denote the meta-parameters as $\phi$, and the pre- and post-adapt inner parameters as $\theta$ and $\theta^K$, respectively. The meta objective as $J^{\text{Out}}$, the inner loop objective as $J^{\text{In}}$. The general objective for GMRL is to maximise the meta objective $J^{\text{Out}}(\phi, \theta^K)$, where the inner-loop post-adapt parameters are obtained by taking $K$ policy gradient steps. Note that all inner-loop updates refer to an expected policy gradient (EPG) and all bias term we discuss hereafter is the bias *w.r.t* the **exact** meta-gradient in this $K$-step EPG inner-loop setting. We discuss the **truncated** $K$-step EPG setting in Appendix B. The form of exact meta-gradient $\nabla_\phi J^K(\phi)$ is given by the following proposition via the chain rule.

**Proposition 3.1** ($K$-step Meta-Gradient). *The exact meta-gradient to the objective in Eq.* (2) *can be written as:*

$$\nabla_\phi J^K(\phi) = \nabla_\phi J^{Out}(\phi, \theta^K) + \alpha \nabla_\phi \theta^K \nabla_{\theta^K} J^{Out}(\phi, \theta^K),$$

$$\nabla_\phi \theta^K = \sum_{i=0}^{K-1} \nabla_\phi \nabla_{\theta^i} J^{In}(\phi, \theta^i) \prod_{j=i+1}^{K-1} \left( I + \alpha \nabla_{\theta^j}^2 J^{In}(\phi, \theta^j) \right).$$

(3)

The derivation of the above proposition is in Appendix E.1.

For each topic mentioned in Sec. 2, we pick one GMRL algorithm example to illustrate how they can be fit into this framework. To describe meta-gradient estimation in RL, we start with basic notations. Consider a discrete-time finite horizon Markov Decision Process (MDP) defined by $\langle \mathcal{S}, \mathcal{A}, p, r, \gamma, H \rangle$. At each time step $t$, the RL agent observes a state $s_t \in \mathcal{S}$, takes an action $a_t \in \mathcal{A}$ based on the policy $\pi_\theta(a_t|s_t)$ parametrised with $\theta \in \mathbb{R}^d$, transits to the next state $s_{t+1} \in \mathcal{S}$ according to the transition function $p(s_{t+1}|s_t, a_t)$ and receives the reward $r(s_t, a_t)$ . We define the return $\mathcal{R}(\tau) = \sum_{t=0}^{H} \gamma^t r(s_t, a_t)$ as the discounted sum of rewards along a trajectory $\tau := (s_0, a_0, \ldots, s_{H-1}, a_{H-1}, s_H)$ sampled by agent policy. The objective for the RL agent is to maximise the expected discounted sum of rewards $V(\theta) = \mathbb{E}_{\tau \sim p(\tau;\theta)}[\mathcal{R}(\tau)]$. Then the RL agent updates parameter $\theta$ using policy gradient given by $\nabla_\theta V(\theta) = \mathbb{E}_{\tau \sim p(\tau;\theta)} [\nabla_\theta \log \pi_\theta(\tau) \mathcal{R}(\tau)]$ where $\nabla_\theta \log \pi_\theta(\tau) = \sum_{t=0}^{H} \nabla_\theta \log \pi_\theta(a_t|s_t)$. For two-agent RL problems, we can extend the MDP to two-agent MDP (or, Stochastic games [31]) defined by $\langle \mathcal{S}_1, \mathcal{S}_2, \mathcal{A}_1, \mathcal{A}_2, P, r_1, r_2, \gamma, H \rangle$, the learning objective for agent $i$ is to maximise its value function of $V_i(\theta) = \mathbb{E}_{\tau_i \sim p(\tau_i;\theta_i)} [\mathcal{R}_i(\tau_i)]$.

**MAML.** Finn et al. [10] optimized over meta initial parameters to maximize the return of one-step adapted policy: $\theta^1 = \theta + \alpha \nabla_\theta V(\theta)$. In MAML-RL, $J^{\text{Out}}(\phi, \theta^1)$ degenerates to $V(\theta^1)$ and $\phi$ and $\theta$ represent the same initial parameters. The meta-gradient can be derived in the form of Eq. (3): $\nabla_\theta \theta^1 \nabla_{\theta^1} V(\theta^1)$, where $\nabla_\theta \theta^1 = I + \alpha \nabla_\theta^2 V(\theta)$.

**LOLA.** Foerster et al. [13] proposed a new learning objective by including an additional term that accounts for the impact of the ego policy to the anticipated opponent's gradient update. For LOLA-agent with parameters $\phi$, it will optimise its return over one-step-lookahead opponent parameters $\theta^1$. The meta-gradient can be shown as: $\nabla_\phi V_1(\phi, \theta^1) + \nabla_\phi \theta^1 \nabla_{\theta^1} V_1(\phi, \theta^1)$, where $\nabla_\phi \theta^1 = \alpha \nabla_\phi \nabla_\theta V_2(\phi, \theta)$

**MGRL.** Xu et al. [39] proposed to tune the discount factor $\gamma$ and bootstrapping parameter $\lambda$ in an online manner. The main feature of MGRL is to conduct inner-loop RL policy $\theta$ update and outer-loop meta parameters $\phi = (\gamma, \lambda)$ alternately. In MGRL, $J^{\text{Out}}(\phi, \theta^1)$ degenerates to $V(\theta^1)$. The meta-gradient takes the form: $\nabla_\phi \theta^1 \nabla_{\theta^1} V(\theta^1)$, $\nabla_\phi \theta^1 = \alpha \nabla_\phi \nabla_\theta V(\phi, \theta)$.

**LPG.** Oh et al. [26] aimed to learn a neural network based RL algorithm, by which a RL agent can be properly trained. In LPG, $\theta$ represents the RL agent policy parameters and $\phi$ is the meta-parameter of neural LSTM RL algorithm, $J^{\text{In}}(\phi, \theta)$ denotes $f(\phi, \theta)$, which is the output of meta-network $\phi$ for conducting inner-loop neural policy gradients. The meta-gradient can be shown as: $\nabla_\phi \theta^1 \nabla_{\theta^1} V(\theta^1)$, $\nabla_\phi \theta^1 = \alpha \nabla_\phi \nabla_\theta f(\phi, \theta)$.

**Remark.** The analytical form of exact meta-gradient given in Eq. (3) involves computation of first-order gradient $\nabla_\theta J^{\text{In}}$, $\nabla_\phi J^{\text{Out}}$ and $\nabla_\theta J^{\text{Out}}$, Jacobian $\nabla_\phi \nabla_\theta J^{\text{In}}$ and Hessian $\nabla_\theta^2 J^{\text{In}}$. In practice,

these four quantities can be estimated by random samples from inner-loop update step $0$ to $K - 1$, which denoted by $\tau_0^{0:K-1}, \tau_1^{0:K-1}, \tau_2^{0:K-1}, \tau_3$. As a result, the estimated gradient $\nabla_\phi \hat{J}^K(\phi)$ can be derived as:

$$\nabla_\phi \hat{J}^K(\phi) = \nabla_\phi \hat{J}^{\text{Out}}(\phi, \hat{\theta}^K, \tau_3) + \alpha \nabla_\phi \hat{\theta}^K \nabla_{\hat{\theta}^K} \hat{J}^{\text{Out}}(\phi, \hat{\theta}^K, \tau_3),$$

$$\nabla_\phi \hat{\theta}^K = \sum_{i=0}^{K-1} \nabla_\phi \nabla_{\hat{\theta}^i} J^{\text{In}}(\phi, \hat{\theta}^i, \tau_1^i) \prod_{j=t+1}^{K-1} \left( I + \alpha \nabla_{\hat{\theta}^j}^2 J^{\text{In}}(\phi, \hat{\theta}^j, \tau_2^j) \right). \tag{4}$$

The post-adapt inner parameter estimate takes the form $\hat{\theta}^K = \theta^0 + \alpha \sum_{i=0}^{K-1} \nabla_{\hat{\theta}^i} \hat{J}^{\text{In}}(\phi, \hat{\theta}^i, \tau_0^i)$.

# 4 Theoretical Analysis of Meta-gradient Estimators

In this section, we systematically discuss and theoretically analyse the bias and variance terms for meta-gradient estimations in the current GMRL literature. We highlight two important sources of biases in meta-gradient estimations: the compositional bias and the multi-step Hessian bias. Our analysis builds on the following three assumptions:

**Assumption 4.1** (Lipschitz continuity). The outer-loop objective function $J^{\text{Out}}$ satisfies that $J^{\text{Out}}(\cdot, \theta)$ and $J^{\text{Out}}(\phi, \cdot)$ are Lipschitz continuous with constants $m_\theta$ and $m_\phi$ respectively, $m_1 = \max_\theta m_\theta$, $m_2 = \max_\phi m_\phi$. $\nabla_\theta J^{\text{Out}}(\cdot, \theta)$ and $\nabla_\theta J^{\text{Out}}(\phi, \cdot)$ are Lipschitz continuous with constants $\mu_\theta$ and $\mu_\phi$ respectively, $\mu_1 = \max_\theta \mu_\theta$, $\mu_2 = \max_\phi \mu_\phi$. The inner-loop objective function $J^{\text{In}}$ satisties that $\nabla_\theta J^{\text{In}}(\cdot, \theta)$ and $\nabla_\theta J^{\text{In}}(\phi, \cdot)$ are Lipschitz continuous with constants $c_\theta$ and $c_\phi$ respectively, $c_1 = \max_\theta c_\theta$, $c_2 = \max_\phi c_\phi$. $\nabla_\theta^2 J^{\text{In}}(\phi, \cdot)$ is Lipschitz continuous with constants $\rho_\phi$, $\rho_2 = \max_\phi m_\phi$.

**Assumption 4.2** (Bias of estimators). Outer-loop stochastic gradient estimator $\nabla_\phi \hat{J}^{\text{Out}}(\phi, \theta, \tau)$ and $\nabla_\theta \hat{J}^{\text{Out}}(\phi, \theta, \tau)$ are unbiased estimator of $\nabla_\phi J^{\text{Out}}(\phi, \theta)$ and $\nabla_\theta J^{\text{Out}}(\phi, \theta)$. Inner-loop stochastic gradient estimator $\nabla_\theta \hat{J}^{\text{In}}(\phi, \theta, \tau)$ is unbiased estimator of $\nabla_\theta J^{\text{In}}(\phi, \theta)$.

**Assumption 4.3** (Variance of estimators). The outer-loop stochastic gradient estimator $\nabla_\phi \hat{J}^{\text{Out}}(\phi, \theta, \tau)$ and $\nabla_\theta \hat{J}^{\text{Out}}(\phi, \theta, \tau)$ has bounded variance, i.e., $\mathbb{E}_\tau[\|\nabla_\phi \hat{J}^{\text{Out}}(\phi, \cdot, \tau) - \nabla_\phi J^{\text{Out}}(\phi, \cdot)\|^2] \le (\sigma_1)^2$, and $\mathbb{E}_\tau[\|\nabla_\theta \hat{J}^{\text{Out}}(\phi, \cdot, \tau) - \nabla_\theta J^{\text{Out}}(\phi, \cdot)\|^2] \le (\sigma_2)^2$.

The above three assumptions are all common ones adopted by existing work [6, 7, 18]. We futher discussed the limitation of assumptions, which are presented in Appendix D.

## 4.1 The Compositional Bias

Recall the $K$-step inner-loop meta-gradient estimate in Eq. (4), existing GMRL methods usually get unbiased outer-loop gradient estimator $\nabla_\phi \hat{J}^{\text{Out}}(\phi, \hat{\theta}^K, \tau_3)$ and $\nabla_{\hat{\theta}^K} \hat{J}^{\text{Out}}(\phi, \hat{\theta}^K, \tau_3)$, then the algorithm can get unbiased meta-gradient estimation by plugging in unbiased inner-loop gradient estimation $\hat{\theta}^K$, where $\mathbb{E}[\hat{\theta}^K] = \theta^K$. However, this is not true because of the compositional optimisation structure.

Consider a non-linear compositional scalar objective $f(\theta^K)$, the gradient estimation bias comes from the fact that

$$f(\theta^K) = f(\mathbb{E}[\hat{\theta}^K]) \ne \mathbb{E}[f(\hat{\theta}^K)].$$

If one substitutes the non-linear function $f(\theta^K)$ with $\nabla_{\theta^K} J^{\text{Out}}(\phi, \theta^K)$ and $\nabla_\phi J^{\text{Out}}(\phi, \theta^K)$, then a typical meta-gradient estimation in GMRL introduces compositional bias:

$$\mathbb{E}[\nabla_{\hat{\theta}^K} \hat{J}^{\text{Out}}(\phi, \hat{\theta}^K, \tau_3)] = \mathbb{E}[\nabla_{\hat{\theta}^K} J^{\text{Out}}(\phi, \hat{\theta}^K)] \ne \nabla_{\theta^K} J^{\text{Out}}(\phi, \theta^K),$$

$$\mathbb{E}[\nabla_\phi \hat{J}^{\text{Out}}(\phi, \hat{\theta}^K, \tau_3)] = \mathbb{E}[\nabla_\phi J^{\text{Out}}(\phi, \hat{\theta}^K)] \ne \nabla_\phi J^{\text{Out}}(\phi, \theta^K), \tag{5}$$

which leads to meta-gradient estimation bias. The following lemma characterises compositional bias.

**Lemma 4.4** (Compositional Bias). *Suppose that Assumption 4.1 and 4.2 hold, let $\hat{\Delta}_C = \mathbb{E}[\|f(\hat{\theta}^K) - f(\theta^K)\|]$ be the compositional bias and $C_0$ the Lipschitz constant of $f(\cdot)$, $|\tau|$ denote number of trajectories used to estimate inner-loop gradient in each inner-loop update step, $\alpha$ the learning rate, then we have,*

$$\hat{\Delta}_C \le C_0 \mathbb{E}[\|\hat{\theta}^K - \theta^K\|] \le C_0 \left( (1 + \alpha c_2)^K - 1 \right) \frac{\hat{\sigma}_{In}}{c_2 \sqrt{|\tau|}}, \tag{6}$$

*where $\hat{\sigma}_{In} = \max_i \sqrt{\mathbb{V}[\nabla_{\theta^i} \hat{J}^{In}(\phi, \theta^i, \tau_0^i)]}$, $i \in \{0, .., K-1\}$.*

*Proof.* See Appendix F.1 for a detailed proof. □

Lemma 4.4 indicates that the compositional bias comes from the inner-loop policy gradient estimate, concerning the learning rate $\alpha$, the sample size $|\tau|$ and the variance of policy gradient estimator $\hat{\sigma}_{\text{In}}$. This is a fundamental issue in many existing GMRL algorithms [13, 39] since applying stochastic policy gradient update can introduce estimation errors, possibly due to large sampling variance, therefore $\hat{\theta}^K \neq \theta^K$. It also implies that the bias issue becomes more serious under the multi-step formulation since each policy gradient step introduces estimation error, resulting in composite biases.

## 4.2 The Multi-step Hessian Bias

Recall the analytical form of the exact meta-gradient in Eq. (3), estimating $\nabla_\phi \theta^K$ involves computing Hessian $\nabla_{\theta^j}^2 J^{\text{In}}(\phi, \theta^j)$. In Eq. (4), the Hessian term is estimated by $\nabla_{\hat{\theta}^j}^2 J^{\text{In}}(\phi, \hat{\theta}^j, \tau_2^j)$ where $j \in \{1, \ldots, K-1\}$. Hessian estimation is a non-trivial problem in GMRL, biased Hessian estimation issue has been brought up in various MAML-RL papers[23, 29, 33], we offer a brief summary in Appendix C due to page limit. Beyond MAML-RL, many recent GMRL work suffers from the same bias due to applying direct automatic differentiation. For example, existing work such as Eq. (3) in Oh et al. [26], Eq. (4) in Zheng et al. [43], Eq. (12), (13), (14) in Xu et al. [39] and Eq. (5), (6), (7) in Xu et al. [40] suffers from this issue. Interestingly, most of them are **coincidentally** unbiased if they only conduct only one-step policy gradient update in the inner-loop. For $K$-step GMRL when $K = 1$, $\nabla_\phi \theta^1$ in meta-gradient writes as:

$$\nabla_\phi \theta^1 = \nabla_\phi \nabla_{\theta^1} J^{\text{In}}(\phi, \theta^1). \tag{7}$$

We can see from Eq. (7) that it would not involve Hessian $\nabla^2 J^{\text{In}}$ computation if $\phi \neq \theta$. To further illustrate, in one-step MGRL, we can show that the estimation of $\nabla_\phi \theta^1$ are unbiased because it takes derivatives *w.r.t* meta-parameters $\phi = (\gamma, \lambda)$ which don't have gradient dependency on the trajectory distribution. However, when it takes more than one-step inner-loop policy gradient updates, the meta-gradient estimation will get the hessian estimation bias. As a result, for the reason that when $K > 1$ in $K$-step GMRL, the $\nabla_\phi \theta^K$ in meta-gradient takes the form:

$$\nabla_\phi \theta^K = \sum_{i=0}^{K-1} \nabla_\phi \nabla_{\theta^i} J^{\text{In}}\left(\phi, \theta^i\right) \prod_{j=t+1}^{K-1} \left(I + \alpha \nabla_{\theta^j}^2 J^{\text{In}}\left(\phi, \theta^j\right)\right). \tag{8}$$

This is the reason why we name it by **multi-step** Hessian bias.

## 4.3 Theoretical Bias-Variance Analysis

Based on Lemma 4.4 and the discussion in Sec. 4.2, we can derive the upper bound on the bias and variance of the meta-gradient with $K$-step inner-loop updates.

**Theorem 4.5** (Upper bound for the bias and the variance). *Suppose that Assumption 4.1 and 4.2 and 4.3 hold. Let $J_{\phi,\theta}$ denote $\nabla_\phi \nabla_\theta J^{In}$, $H_{\theta,\theta}$ denote $\nabla_\theta^2 J^{In}$, $\hat{\Delta}^K = \|\mathbb{E}[\nabla_\phi \hat{J}^K(\phi)] - \nabla_\phi J^K(\phi)\|$ be the meta-gradient estimation bias, set $B = 1 + \alpha c_2$. Then the bound of bias hold:*

$$\hat{\Delta}^K \leq O\left((B + \hat{\Delta}_H)^{K-1}\left(\mathbb{E}[\|\hat{\theta}^K - \theta^K\|] + \hat{\Delta}_J + (K-1)\right)\right). \tag{9}$$

*Let $(\hat{\sigma}^K)^2 = \mathbb{V}\left[\nabla_\phi \hat{J}^K(\phi)\right]$ be the meta-gradient estimation variance, set $V_1 = (1 + \alpha c_2)^2$, $V_2 = 2\alpha^2(m_1^2 + 3\sigma_2^2)$ the estimation variance is given by*

$$(\hat{\sigma}^K)^2 \leq O\Bigg((V_1 + \hat{\Delta}_H^2)^{K-1}\left(\mathbb{E}[\|\hat{\theta}^K - \theta^K\|^2] + (K-1)\right)$$

$$+ \left(V_2 + (V_1 + \hat{\Delta}_H^2 + \hat{\sigma}_H^2)^{K-1} - (V_1 + \hat{\Delta}_H^2)^{K-1}\right)(\hat{\Delta}_J^2 + \hat{\sigma}_J^2)\Bigg). \tag{10}$$

*where $\hat{\Delta}_J = \max_{\phi \times \theta} \|\mathbb{E}[\hat{J}_{\phi,\theta}] - J_{\phi,\theta}\|$. $\hat{\Delta}_H = \max_\theta \|\mathbb{E}[\hat{H}_{\theta,\theta}] - H_{\theta,\theta}\|$. $(\hat{\sigma}_J)^2 = \frac{\max_{\phi \times \theta} \mathbb{V}[\hat{J}_{\phi,\theta}]}{|\tau|}$. $(\hat{\sigma}_H)^2 = \frac{\max_\theta \mathbb{V}[\hat{H}_{\theta,\theta}]}{|\tau|}$.*

*Proof.* See Appendix G.1 for a detailed proof. □

Theorem 4.5 shows that the upper bound of bias and variance consists of two parts: the first term indicates compositional bias $\mathbb{E}[\|\hat{\boldsymbol{\theta}}^K - \boldsymbol{\theta}^K\|]$ in Lemma 4.4, while the second term refers to the second-order estimation bias $(\hat{\Delta}_J, \hat{\Delta}_H)$ and the variance $(\hat{\sigma}_J, \hat{\sigma}_H)$. Several observations can be made based on Theorem 4.5: 1) the compositional bias exists in both upper bound of bias and variance; 2) the multi-step Hessian bias has polynomial impact on the upper bound of bias and variance; 3) most importantly, many existing GMRL algorithms suffer from the compositional bias; moreover, the Hessian bias can significantly increase meta-gradient bias in the multi-step inner-loop setting.

## 4.4 Understanding Existing Mitigations for Meta-gradient Biases

Based on the above theoretical analysis, we now try to explore and understand how existing methods can handle these two estimation biases.

**Off-policy Learning.** From the Lemma 4.4 and the discussion in Sec. 4.1, we know that the compositional bias is caused by the estimation error between $\hat{\theta}_t^K$ and $\theta_t^K$. In this case, a simple idea is that one can leverage off-policy learning technique [32] to handle the compositional bias problem by reusing samples $\tau_{0:t-1,0}^i$ together with $\tau_{t,0}^i$ to approximate $\theta_t^K$, we want $\hat{\theta}_t^K$ to stay close to $\theta_t^K$. The intuition behind this correction is to enlarge the sample size $|\tau|$ so that the compositional bias can be minimised according to Lemma 4.4. Specifically, we can apply importance sampling technique to correct the compositional bias, i.e., $\mathbb{E}_{\tau \sim p(\tau;\theta)} \left[ \sum_{t=0}^{H-1} \frac{\pi_\theta(a_t|s_t)}{\mu(a_t|s_t)} \mathcal{R}_\phi(\tau) \right]$. In practice, we also need to manually control the level of off-policyness to prevent the variance introduced by the importance sampling process from increasing the estimation variance conversely.

**Multi-step Hessian Estimator.** From the theoretical analysis in Sec. 4.3, we can know that the Hessian estimation bias can significantly increase the meta-gradient estimation bias in the multi-step inner-loop setting. Many low-bias hessian estimator have been proposed in the scope of MAML-RL. However, the effect of them have never been verified in the general GMRL context. Here we have a second look at the Low Variance Curvature (LVC) method [29], one can replace the original log-likelihood with the LVC operator in the policy gradient step. As such, the Hessian estimator $\nabla_\theta^2 J_{\text{LVC}}^{\text{In}}(\phi, \theta) =$ takes the form: $\nabla_\theta \mathbb{E}_{\tau \sim p(\tau;\theta)} \left[ \sum_{t=0}^{H-1} \frac{\nabla_\theta \pi_\theta(a_t|s_t)}{\perp \pi_\theta(a_t|s_t)} \mathcal{R}_\phi(\tau) \right]$ where $\perp$ is the stop-gradient operation which detaches the gradient dependency from the computation graph. As shown in [29], the LVC operator will ensure an unbiased first-order policy gradient and low-biased low-variance second-order policy gradient. Essentially, this operator only corrects the meta-gradient update and leaves the inner-loop gradient estimation formula untouched.

## 5 Experiments

In this section we conduct empirical evaluation of our proposed bias analysis and the proposed methods to mitigate the biases, and our experiments cover all 3 GMRL fields listed in Table 1. In particular, we conduct a tabular MDP experiment to show the existence of two biases discussed in Sec. 4 using MAML [10] and LIRPG [42]. In order to show how the proposed methods can mitigate the compositional bias, we consider a Iterated Prisoner Dilemma (IPD) problem and use LOLA [13] with off-policy corrections in Sec. 4.4; Similarly, we conduct evaluations on Atari games using MGRL [39] with LVC corrections in Sec. 4.4 to mitigate the Hessian estimation bias. We open source our code at https://github.com/Benjamin-eecs/Theoretical-GMRL.

### 5.1 Investigate the correlation and bias of meta-gradient estimators

Firstly, we conduct experiments to study different meta-gradient estimators using a tabular example using MAML and LIRPG. To align with existing works in the literature, we adopt the settings of random MDPs in [33] with the focus on meta-gradient estimation. We refer readers to Appendix I.1.1 for more experimental details.

**MAML-RL.** Recall that the meta-gradient in MAML is $\nabla_{\theta^0} J^{\text{Out}}(\theta^K)$. To control the effect of gradient estimations, we use the three estimators to estimate following three terms: (I) inner-loop policy gradient $\nabla_\theta J^{\text{In}}(\theta)$; (II) Jacobian/Hessian $\nabla_\theta^2 J^{\text{In}}(\theta)$; (III) outer-loop policy gradient $\nabla_{\theta^K} J^{\text{Out}}(\theta^K)$. Refer to Appendix I.1.2 for the implementation of decomposing meta-gradient estimation with different estimators.

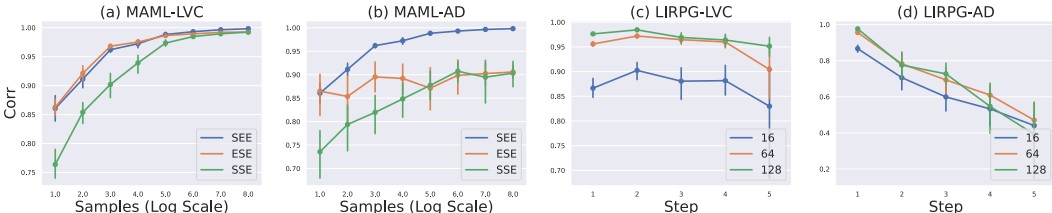

Figure 1: (a, b) Ablation study on sample size and estimators in MAML-RL. "S" is for stochastic estimation while "E" is for exact solution. AD refers to automatic differentiation. (c,d) Ablation study on sample size, steps and estimators in LIRPG.

To show how the estimation is biased, we use a stochastic estimator denoted as S, and exact analytic calculator denoted as E, for all three derivative terms. Thus, we can have 7 valid permutations in the experiment to validate the estimation, where the rest EEE estimator is the exact gradient.

**Firstly, we conduct our ablation studies by comparing the correlation between of meta-gradient with the exact one.** The correlation metric, which is determined by bias and variance, can show how the final estimation quality is influenced by these two bias terms. As the quality of stochastic estimators vary from many factors, we conduct this ablation study under extensive combinations of estimation algorithms (including DiCE [12], Loaded-DiCE [8], LVC [29], and pure automatic differentiation in original MAML [10]), learning rates, sample sizes, .etc. Due to the page limit, the results illustrated in Fig. 1(a,b) only include ablation study on sample size using LVC/automatic differentiation and estimation using exact gradients for (III), the outer-loop policy gradient. For the rest ablation study and more evaluation metrics (variance of estimation), refer to Appendix I.1.3.

We start our evaluation by increasing the sample size. For simplicity, we only conduct one inner step here. In Fig. 1(a), we leverage the LVC [29] estimator. We can see that a correct inner-loop policy estimation and/or a Hessian estimation can significantly improve the estimation quality (as SEE ≈ ESE > SSE in low sample size case). By increasing the sample size, we can see the gap between them is shrinking, which verifies the finding in Lemma 4.4. In this case the compositional bias correction shares the same importance with the Hessian bais correction (SEE ≈ ESE). We also compare it with the original yet biased gradient estimator of MAML in Fig. 1(b), in which SEE > ESE in all sample size settings. In fact, since the gradient estimation is biased, only SEE achieves near 1.0 correlation provided sufficient samples, again confirms the importance of Hessian bias corrections in Sec. 4.2.

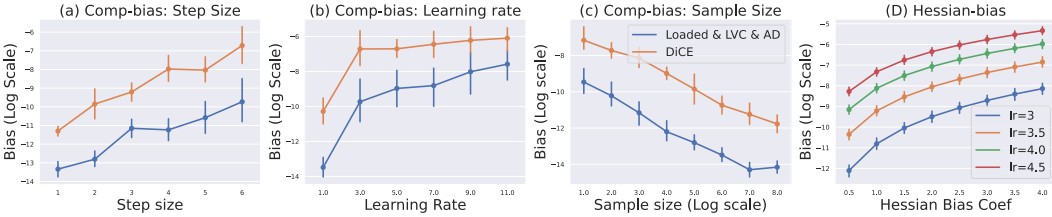

Figure 2: (a, b, c) Ablation study of meta-gradient bias due to the compositional bias in different estimators, step sizes, learning rates. Loaded-DiCE, LVC and AD achieve exactly the same compositional bias because they have the same first-order gradient, (d) Ablation study of meta-gradient bias due to the Hessian bias in different learning rates and Hessian bias coefficients.

**Beside the correlation result above, we also add additional experimental results in Fig. 2 over the pure meta-gradient bias term introduced by compositional bias and Hessian bias.** It can also be regarded as an empirical verification of our Lemma 4.4 and Theorem 4.5. In Fig. 2 (a, b, c), we mainly study How (a) the inner-loop step size, (b) learning rate and (c) sample size influence final meta-gradient bias. It successfully validates our Lemma 4.4 ($O(K\alpha^K \hat{\sigma}_{\text{In}} |\tau|^{-0.5})$) about the exponential impact from the inner-loop step $K$ (approximately linear relationship between log-scale bias and step size in (a)), the polynomial impact from the learning rate $\alpha$ (approximately Concave downward relationship between log-scale bias and learning rate in (b)) and the polynomial impact

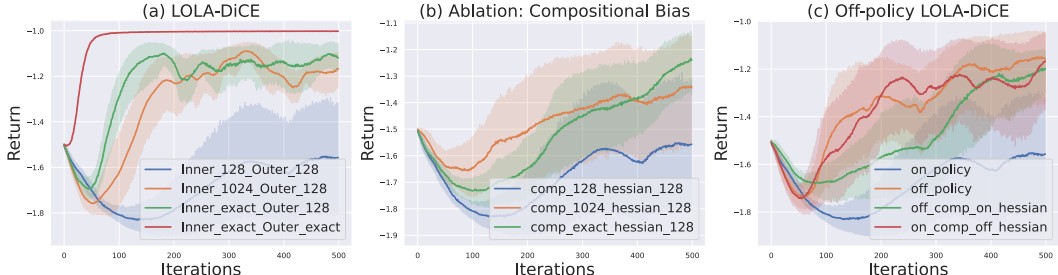

Figure 3: Experiment result of LOLA-DiCE over 10 seeds. The Inner_*A*_Outer_*B* legend means we use *A* samples to estimate inner-loop gradient while *B* samples to estimate outer-loop gradient. The 'exact' means we use analytical solution of policy gradient instead of estimation.

from the sample size $\alpha$ (approximately negative linear relationship between log-scale bias and log-scale sample size in (c)). In the second MAML-Hessian experiment, we conduct experiments to verify the polynomial impact $O\big((K-1)(\hat{\Delta}_H)^{K-1}\big)$ on the meta-gradient bias introduced by the multi-step Hessian estimation bias $\hat{\Delta}_H$ (The Concave downward relationship in (d)). In our implementation, we manually add the Hessian bias error into the estimation and control the quantity of it by multiplying different coefficients.

**LIRPG.** In this setting, we follow the algorithm of intrinsic reward generator presented in [42]. In tabular MDP, we have an additional meta intrinsic reward matrix $\phi$. Starting from $\theta^0$, the inner-loop process takes policy gradient based on the new reward matrix $R_{\text{new}} = R + \phi$: $\theta^{i+1} = \theta^i + \alpha \nabla_{\theta^i} J^{\text{In}}(\theta^i, \phi), i \in \{0, 1...K-1\}$. The meta-gradient estimation of the intrinsic reward matrix $\nabla_\phi J^{\text{Out}}(\theta^K)$ is needed in this case. Note that in the outer loss we use the original reward matrix $R$ so the outer loss is $J^{\text{Out}}(\theta^K)$ rather than $J^{\text{Out}}(\phi, \theta^K)$. Compared with MAML-RL, the object of meta-update (intrinsic matrix) and the object of inner-update (policy parameters) are different, which help us identify the problem mentioned in Sec. 4.2.

In this case, we choose the LVC and AD estimator. We conduct ablation study on inner-step and sample size shown in Fig. 1(c,d). With more sample size and less step size, the correlation increases for both estimator. Two important features are: (1) With 1-step inner-loop setting, both estimator performs similarly in the correlation. (2) With multi-step inner-loop setting, LVC based estimator can still reach relatively high correlation while MAML-biased estimator directly reaches low correlation after 5-step inner-loop. The phenomenon shown here corresponds exactly to the Hessian estimation issue we discuss in Sec. 4.2 and the bias issue will be more severe with multi-step inner-loop setting.

## 5.2 Compositional bias/off-policy learning in LOLA

In this subsection, we conduct three experiments on Iterated Prisoner Dilemma (IPD) with the LOLA algorithm to show: (1) The effect brought by different inner/outer estimators. (2) The effect brought compisitional bias (Sec. 4.1) (3) How off-policy correction (Sec. 4.4) can help the LOLA algorithm. Please refer to Appendix I.2.1, I.2.2 for more experimental setting and results.

**Ablation on LOLA-DiCE inner/outer estimation.** We report the result of conducting ablation study for different inner/outer-loop estimation of LOLA-DiCE in the Fig. 3(a). Here the inner-loop estimation refers to $\nabla_\theta J^{\text{In}}(\phi, \theta)$ while outer-loop estimation refers to $\nabla_{\theta^1} J^{\text{Out}}(\phi, \theta^1)$ and $\nabla_\phi J^{\text{Out}}(\phi, \theta^1)$. The return shown in Fig. 3(a) reveals us two findings: 1) The inner-loop gradient estimation plays an important role for making LOLA work—the default batch size 128 fails while the batch size 1024 succeeds. 2) The outer-loop gradient estimation is also crucial to the performance of LOLA-Exact. Furthermore, we continue conducting ablation studies on the inner-loop gradient update.

**Ablation on compositional bias.** Since the unbiased DiCE estimator is used in LOLA-DiCE algorithm, there is no Hessian estimation bias in the LOLA algorithm. Thus, we mainly discuss the problem of compositional bias brought in Fig. 3(b). We also apply the implementation in Sec. 5.1 to decompose meta-gradient estimation with different estimators. Fig. 3(b) show us the ablation study over compositional bias, which reveals that: compositional bias may decrease the performance and by adding more samples or using analytical solution, the performance can start to improve.

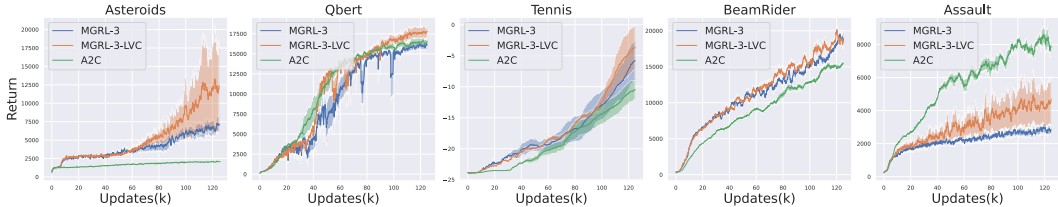

Figure 4: Experimental results on Atari game over 5 random seeds.

**Off-policy DiCE and ablation study.** We use the off-policy learning to conduct inner-loop update and keep the outer-loop gradient same as before. By combing DiCE and off-policy learning, we have off-policy DiCE $J^{\text{OFF−DICE}}$: $\mathbb{E}_\tau \left[ \sum_{t=0}^{H-1} \left( \prod_{t'=0}^{t} \frac{\pi_\phi(a_{t'}^1|s_{t'}^1)\pi_\theta(a_{t'}^2|s_{t'}^2)}{\mu_1(a_{t'}^1|s_{t'}^1)\mu_2(a_{t'}^2|s_{t'}^2)} \right) R_t \right]$, where $\phi, \theta$ refer to the current policy, $\mu_1, \mu_2$ refer the behaviour policy for agent 1 and agent 2, respectively. H is the trajectory length and $R$ refers to the reward for agent. Note that the off-policy DiCE here can not only lower the compositional bias by lowering the first-order policy gradient error, but also helps lower the Hessian variance theoretically. By the decomposition trick, we conduct experiments by traversing over all learning settings for (off-off/off-on/on-off/on-on), which are shown at Fig. 3(d). Comparisons between different settings verify that off-policy DiCE can increase performance by either lowering the compositional bias, or the hessian variance, or both.

### 5.3 Multi-step Hessian correction on MGRL

Finally, we conduct experiment over MGRL [39]. See Appendix I.3.1 and I.3.2 for experimental settings. When applying the LVC estimator in MGRL, we get the new inner-loop update equation: $\mathbb{E}_{\tau \sim p(\tau;\theta)} \left[ \sum_{t=0}^{H-1} \frac{\nabla_\theta \pi_\theta(a_t|s_t)}{\perp \pi_\theta(a_t|s_t)} \left( g_\phi(\tau) - v_\theta(s_t) \right) \right]$, where $\phi = (\gamma, \lambda)$ refer to meta-paramters and $\theta$ refers to the RL policy, $g_\phi(\tau)$ denotes $\lambda$-return, $v_\theta(s_t)$ denotes value prediction. We conduct experiment on eight environments of Atari games. We follow previous work [4] to use the "discard" strategy in which we conduct multiple virtual inner-loop updates for meta gradient estimation. This strategy is designed to keep the inner learning update unchanged.

In Fig. (4), we show five environments comparing three variants of algorithm: 1) Baseline Advantage Actor-critic(A2C) algorithm [25]; 2) 3-step MGRL + A2C; 4) 3-step MGRL + A2C + LVC correction. The "3-step" means we take 3 inner-loop RL virtual updates for calculating meta-gradient. Refer to Appendix I.3.3 for experimental results on all eight environments. Compared with 3-step MGRL, the MGRL with LVC correction can substantially improves the performance, which validates the effectiveness of the multi-step Hessian correction in Sec. 4.4 for handling meta-gradient estimation bias and bring in better hyperparameter-tuning in RL. Note that the fact that A2C algorithms can achieve better results compared with 3-step MGRL is consistent with the results in [39].

## 6 Conclusion

In this paper, we introduce a unified framework for studying generic meta-gradient estimations in gradient-based Meta-RL. Based on this framework, we offer two theoretical insights that **1)** the compositional bias has an upper bound of $O(K\alpha^K \hat{\sigma}_{\text{In}}|\tau|^{-0.5})$ with respect to the inner-loop update step $K$, the learning rate $\alpha$, the estimate variance $\hat{\sigma}_{\text{In}}^2$ and the sample size $|\tau|$, and **2)** the multi-step Hessian bias $\hat{\Delta}_H$ has a polynomial impact of $O((K-1)(\hat{\Delta}_H)^{K-1})$. To validate our theoretical discoveries, we conduct a comprehensive list of ablation studies. Empirical results over tabular MDP, LOLA-DiCE and MGRL validate our theories and the effectiveness of correction methods. We believe our work can inspire more future work on unbiased meta-gradient estimations in GMRL.

## Acknowledgements

We would like to thank Yunhao Tang for insightful discussion and help for tabular experiments.

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
