# OpenReview forum: "A Theoretical Understanding of Gradient Bias in Meta-Reinforcement Learning"
_NeurIPS.cc/2022/Conference — NeurIPS 2022 Accept_

### Official Review · Reviewer_RjbJ · 2022-06-30

**Rating:** 5
**Confidence:** 4
**Soundness:** 3 good
**Presentation:** 2 fair
**Contribution:** 2 fair

**Summary:**

The paper comprehensively investigates the gradient bias in Meta-Reinforcement Learning. The authors point out two source of bias: the compositional bias and the multi-step Hessian bias. The paper then gives a generic bound on the bias and variance of the final meta-gradient. The paper mitigates the two biases using off-policy training and LVC estimator respectively. A debased algorithm is tested on 8 Atari games.

**Questions:**

Line 52: not clear what does the impact mean here
Line 204 typo: an extra "the"
Line 216: It seems there are multiple bias terms in (9). Why is $\hat \Delta_J$ less important than $\Delta_C$ and $\Delta_H$? It would be better to use $\Delta_C$ instead of $E[\|\hat \theta^K - \theta^K\|]$ here as it has already been defined.
Line 218: What does the variance term imply here? Is there any algorithm in the literature that can balance bias and variance? How well do they work in the decomposition in (9)?
Line 219. The use of punctuation is bad here. What set is the maximization taken over?
Line 318. What is a DiCE estimator?


**Limitations:**

The authors did not address any potential negative societal impact. As a theoretical paper, I don't think it is necessary.

**Strengths And Weaknesses:**

Strengths:
I believe the paper is significant in the following ways
1. This is one of the first paper to give a generic analysis on gradient bias, which applies to most of the algorithms.
2. The GMRL framework proposed in the paper applies to a number of important meta-RL frameworks.
3. The analysis that decomposes the bias terms is novel and interesting.

Weakness:
1. I am concerned about the actual contributions of the paper. The paper provide theoretical analysis that decompose the gradient bias into several terms. However,  the experiments are only weakly dependent on the theories.

To be specific, the paper adapts off-policy training to mitigate the compositional bias. Enlarging the number of data to mitigate bias is a trivial solution opposed to actual bias correction. As the paper mentioned, off-policy training suffers the shifts in the data collection policies and importance sampling process has to be used. LVC is a method existing in the literature. Thus, the paper is not able to propose a novel algorithm that actually balances the bias and variance based on Theorem 4.5 to achieve a better performance.

The experiments in 5.1 simply show that the performance is higher when replacing sample-based estimator with exact solution, which is quite obvious in my opinion.

The paper does not clearly state the main contribution over the literature. There are papers [33, 6] discussing bias in meta-RL. I believe [6] also considers the compositional bias. I don't understand the main contribution here except for proposing a more general framework that applies to different cases.

2. The paper is not well-organized and it can be hard to follow. Different sections are not strongly connected.

---

> ### Author Response · Authors · 2022-08-02
> **Response to Reviewer RjbJ (1)**
>
> Many thanks for the most helpful review! Based on the review comments, we have rewritten Lemma 4.4 and Section 4.2, via which we hope that we have addressed the reviewer’s previous concerns satisfactorily. We will appreciate it very much if the reviewer can go over these revisions and re-evaluate the paper. The updated PDF of our paper has been uploaded.
>
> > Q: The experiments are only weakly dependent on the theories.
>
> According to Lemma 4.4, $\hat{\Delta}\_{C} \leq C_0 \left((1+\alpha c_2)^{K} -1\right) \frac{\hat{\sigma}\_{\text{In}}}{c_2\sqrt{|\boldsymbol{\tau}|}}$, we can conclude that the compositional bias has an upper bound of $\mathcal{O}\big(K\alpha^{K}\hat{\sigma}\_{\text{In}}|\tau|^{-0.5}\big)$ with respect to the inner-loop update step $K$, the learning rate $\alpha$, the estimate variance $\hat{\sigma}^{2}{\text{In}}$ and the sample size $|\tau|$ and we show that in Theorem 4.5 Multi-step Hessian bias grows exponentially as the inner-loop update step increasing and this bias will affect total meta-gradient bias in a polynomial way $\mathcal{O}\big((K-1)(\hat{\Delta}\_{H})^{K-1}\big)$. In our experiment, we investigate how different estimation condition (including extensive combinations of estimation algorithms, learning rates and sample sizes) can influence the final gradient quality, which corresponds to the quantity in our theory. For further experimental results, please refer to the Appendix G.1.1. Based on two theoretical insights gained from Lemma 4.4 and Theorem 4.5: compositional bias can be reduced if we minimize $\mathbb{E} [\|\hat{\boldsymbol{\theta}}^{K}  - \boldsymbol{\theta}^{K}\|]$ and Hessian bias can affect meta-gradient bias greatly with more inner-loop update steps, we come up with off-policy learning to reduce compositional bias and use LVC estimator selected from experiment 5.1 to mitigate Hessian bias on different GMRL settings to show the effectiveness of our methods.
>
> > Q: Off-policy is trivial, LVC is existing method. Thus, the paper is not able to propose a novel algorithm that actually balances the bias and variance based on Theorem 4.5 to achieve a better performance.
>
> We want to clarify that we do not aim at proposing new algorithms, but just to understand investigate how existing methods can work in a completely different use case. Our usage of off-policy learning is to decrease the inner-loop policy gradient variance. But this is different from the normal use case where we aim at sample-efficient RL policy by reusing examples. Our final objective is to decrease the compositional bias and finally get better meta-gradient estimation for meta optimization in RL. The results shown in LOLA-DiCE also validate the effectiveness of this simple method in a completely different use case. For the Hessian bias reduction problem, we choose the existing LVC method. However, it has not been applied out of the scope of MAML-RL and basically most GMRL paper ignores this problem we discuss in Section 4.2 (line 200-201). Thus, we believe even though we use the existing LVC method, our novel use case can also contribute to the community for the awareness of such problem in general GMRL problem.
>
>
> > Q: There are papers [33, 6] discussing bias in meta-RL. I believe [6] also considers the compositional bias. I don't understand the main contribution here.
>
> Our paper consider different objective function (expected update) compared with [33] (K-sample inner-loop update) and we briefly describe it and the differences in line 71-76. We further discuss the setting difference in Appendix B. For [6], it mainly discussed the compositional bias in supervised learning setting, or in specific, MAML where the inner-loop is SGD. We are the first to theoretically analyze this problem and also emprically investigate its effect in the scope of Meta-RL. In RL such problem will be more severe because RL tends to have much larger varaince compared with supervised learning. Plus, the theoretical analysis presented in 4.3 are considering the coupling effect of compositional bias and multi-step hessian bias, which result in the final gradient bias and variance. This is also RL-specific.
>
> > Q: The experiments in 5.1 simply show that the performance is higher when replacing sample-based estimator with exact solution.
>
> Our initial motivation for experiment 1 is to illustrate how different gradient estimation conditions can affect the final meta-gradient estimation. That is why we finally pick correlation for presenting the ablation study. And we also offer the meta-gradient varaince plot in figure 4(Appendix G.1.3), which can further help us with the meta-gradient variance analysis of influence brought by compositional bias and Hessian bias. We are also working on updating the experimental results to further derive the bias plot in different estimation conditions for clearer presentation and stronger connection with our theory.

---

> > ### Author Response · Authors · 2022-08-02
> > **Response to Reviewer RjbJ (2)**
> >
> > > Q: Line 52: what does the impact mean here?
> >
> > We aim to show how the Hessian bias term present in meta-gradient bias. Therefore we use the word 'impact' to describe the effect hessian bias have on total meta-gradient bias.
> >
> > > Q: Line 204 typo: an extra "the"
> >
> > Many thanks! We fixed it in the revision.
> >
> > > Q: Line 216: It seems there are multiple bias terms in Equation (9).  Why is $\hat{\Delta}\_{J}$ less important than $\hat{\Delta}\_{C}$ and $\hat{\Delta}\_{H}$ ?
> >
> > According to the definition of $\hat{\Delta}\_{J}=\max_{\boldsymbol{\phi}\times\boldsymbol{\theta}} \|\mathbb{E}[\hat{J_{\boldsymbol{\phi},\boldsymbol{\theta}}}]-J_{\boldsymbol{\phi}, \boldsymbol{\theta}}\|$, we don't make unbiased estimation assumptions on $J_{\boldsymbol{\phi}, \boldsymbol{\theta}}$, but we can usually get unbiased estimation of $J_{\boldsymbol{\phi}, \boldsymbol{\theta}}$. If $\boldsymbol{\phi} \neq \boldsymbol{\theta}$, taking gradient of $\nabla_{\boldsymbol{\theta}} J^{\text {In}}$ with respect to $\boldsymbol{\phi}$ will not incur the biased Hessian estimation problem mentioned in Appendix C, so $\hat{\Delta}\_{J} =0 $ when $\boldsymbol{\phi} \neq \boldsymbol{\theta}$.  We care $\hat{\Delta}\_{J}$ only if $\boldsymbol{\phi} = \boldsymbol{\theta}$, then $\hat{\Delta}\_{J}$ becomes $\hat{\Delta}\_{H}$ and that's why it is presented in Equation (9).
> >
> >
> > > Q: It would be better to use $\hat{\Delta}\_{C}$ instead of $\mathbb{E} [\|\hat{\boldsymbol{\theta}}^{K}  - \boldsymbol{\theta}^{K}\|]$ here as it has already been defined.
> >
> > Thanks for the great suggestion, we fixed it in the revision. We clarified the difference between $\hat{\Delta}\_{C}$ and $\mathbb{E} [\|\hat{\boldsymbol{\theta}}^{K}  - \boldsymbol{\theta}^{K}\|]$ in the revision, $\mathbb{E} [\|\hat{\boldsymbol{\theta}}^{K}  - \boldsymbol{\theta}^{K}\|]$ is a part of upper bound of $\hat{\Delta}\_{C}$. Therefore we will stick to $\mathbb{E} [\|\hat{\boldsymbol{\theta}}^{K}  - \boldsymbol{\theta}^{K}\|]$ in Theorem 4.5.
> >
> > > Q: Line 218: What does the variance term imply here? Is there any algorithm in the literature that can balance bias and variance? How well do they work in the decomposition in Equation (9)?
> >
> > There exist series of MAML algorithms trying to balance bias and variance, ProMP[2], TayPO-2[3] and TMAML[4]. However, most of them only focus on the bias/variance blance of the Hessian estimation. In our paper, our bias and variance presented in Theorem 4.5 is with resepct to the final meta-gradient. Also, we can find out that the variance bound are also influenced by the compositional and Hessian bias. So here our main concern is not to balance the meta-gradient bias/vairance, but to reduce the compositional bias/Hessian bias. This is helpful for both meta-gradient bias and variance reduction.
> >
> > > Q: Line 219. The use of punctuation is bad here. What set is the maximization taken over?
> >
> > Corresponding argument space. To be more specific, $J_{\boldsymbol{\phi}, \boldsymbol{\theta}}$ is a function of $\boldsymbol{\phi}$ and $\boldsymbol{\theta}$, so we take maximization over cartesian product of parameter space $\boldsymbol{\phi}$ and $\boldsymbol{\theta}$: $\boldsymbol{\phi}\times\boldsymbol{\theta}$ to define $\hat{\Delta}\_{J}$, $\hat{\sigma}\_{J}$ is defined in the similar way. $H_{\boldsymbol{\theta}, \boldsymbol{\theta}}$ is a function of $\boldsymbol{\theta}$, so we take maximization over cartesian product of parameter space $\boldsymbol{\theta}$ to define $\hat{\Delta}\_{H}$, $\hat{\sigma}\_{H}$ is defined in the similar way.
> >
> > > Q: Line 318. What is a DiCE estimator?
> >
> > The DiCE estimator is the infinitely differentiable monte carlo estimator proposed in [1]. It can be used for estimation of any-order RL gradient. In LOLA-DiCE, the DiCE is utilized for the Hessian estimation.
> >
> > [1] Foerster, J., Farquhar, G., Al-Shedivat, M., Rocktäschel, T., Xing, E., & Whiteson, S. (2018, July). Dice: The infinitely differentiable monte carlo estimator. In International Conference on Machine Learning (pp. 1529-1538). PMLR.
> >
> > [2] Rothfuss, J., Lee, D., Clavera, I., Asfour, T., & Abbeel, P. (2018). Promp: Proximal meta-policy search. arXiv preprint arXiv:1810.06784.
> >
> > [3] Tang, Y., Kozuno, T., Rowland, M., Munos, R., & Valko, M. (2021). Unifying gradient estimators for meta-reinforcement learning via off-policy evaluation. Advances in Neural Information Processing Systems, 34, 5303-5315.
> >
> > [4] Liu, H., Socher, R., & Xiong, C. (2019, May). Taming maml: Efficient unbiased meta-reinforcement learning. In International conference on machine learning (pp. 4061-4071). PMLR.

---

> > ### Comment · Reviewer_RjbJ · 2022-08-07
> > **Thanks for the responses**
> >
> > The responses on how the experiments corresponds to the theories address some of my concerns. And the discussions on the literature are also helpful. I decide to raise my score to 5.

---

> > > ### Author Response · Authors · 2022-08-08
> > > **Further update on the result**
> > >
> > > Thank you for your insightful review again. Here we update the experimental  results for the following question.
> > >
> > > > Q: The experiments in 5.1 simply show that the performance is higher when replacing sample-based estimator with exact solution.
> > >
> > > In these days we add more experiments over ths bias term to make the clearer presentation and stronger connection with our theory. We update our result in **Appendix J.1.3 (Figure 5)** and here is a summarization of our additional experiments: In the first experiment, we successfully validate our Lemma 4.4 ($\mathcal{O}\big(K\alpha^{K}\hat{\sigma}\_{\text{In}}|\tau|^{-0.5}\big)$) about the **exponential** impact brought by the inner-loop step $K$ and in the second experiment,  we conduct experiments to verify the **polynomial** impact $\mathcal{O}\big((K-1)(\hat{\Delta}\_{H})^{K-1}\big)$ on the meta-gradient bias introduced by the multi-step Hessian estimation bias $\hat{\Delta}\_{H}$.
> > >
> > > We are also working on further experiments considering ablation study over different estiamtion conditions(learning rate/estimator/sample size) and will update the paper as soon as possible.

---

> ### Author Response · Authors · 2022-08-07
> **Looking forward to further discussion**
>
> Dear Reviewer RjbJ, We really appreciate your insightful comments on our paper. As the discussion period is coming to an end, we really hope you can see if our responses address your concerns or if you have any further questions. Really look forward to further discussing with you about our paper.

---

### Official Review · Reviewer_4HrY · 2022-07-09

**Rating:** 6
**Confidence:** 3
**Soundness:** 3 good
**Presentation:** 3 good
**Contribution:** 2 fair

**Summary:**

This paper studies the bias of gradient-based meta-RL algorithms. The authors provide a bias bound that is made up of a compositional bias and the multi-step Hessian estimation bias, which also suggests ways of understanding current algorithms.

**Questions:**

Could the authors run the experiments for more iterations? I would also like to see if both sources of bias can be combined together or will they suggest new algorithmic designs. Besides, more related work on the theory side is encouraged.

**Limitations:**

Yes.

**Strengths And Weaknesses:**

pros: 1. The paper is well written and easy to follow. Both the background and the analysis are thorough.

2. The bias theorem may also guide future algorithmic designs of meta-RL algorithms, considering where the bias comes from.

3. The experiments also provide evidence supporting the theorem.

cons: 1. In experiments Fig. 2 and Fig. 3, it seems that the curves have not converged.

2. I'm not an expert in meta-RL. Are there other existing works that give bias bounds? I am only aware of the algorithmic designs mentioned in the paper, but I didn't see the related work in the theory aspect.

3. It seems that the authors only evaluate existing modifications to support the theorem instead of proposing new algorithms. Are there ways to combine some of them together to reach the best theoretical and experimental results?

---

> ### Author Response · Authors · 2022-08-02
> **Response to Reviewer 4HrY**
>
> Many thanks for the most helpful review! Based on the review comments, we have added a more extended discussion of prior work on theoretical GMRL in Appendix E, via which we hope that we have addressed the reviewer’s previous suggestions satisfactorily. We will appreciate it very much if the reviewer can go over these revisions and re-evaluate the paper. The updated PDF of our paper has been uploaded.
>
> > Q: In experiments Fig. 2 and Fig. 3, it seems that the curves have not converged.
>
> Regarding Figure 2., we align the training with lola[1] for same number of iterations. As for Figure 3., we use same training setting with lirpg[2] (40M, on 8 games).
>
> > Q: Are there other existing works that give bias bounds? Besides, more related work on the theory side is encouraged.
>
> Due to the highly complex objective landscape of meta learning, most theoretical analysis focuses on convergence to stationary points. Fallah et al.[3] established generic convergence guarantees for gradient-based meta-learning algorithms for supervised learning with one
> inner loop update. Recently, Ji et al.[4] extended the analysis to multi-step inner loop updates. For meta-RL, Fallah et al.[5] established convergence for the EMAML objective. In our paper, we consider a different expected inner-loop update in general Meta-RL problem. And we are the first to theoretically and empirically investigate the compositional bias in RL setting, and the multi-step Hessian bias out of the scope of MAML-RL. We briefly summarize the related theoretical work and describe their difference with our work in line 70-76.
>
> > Q: Are there ways to combine some of them together to reach the best theoretical and experimental results?
>
> What we present in section 4.4 and experiments 5.2/5.3 is to seperately mitigate the bias and we can combine them together to the best bias reduction.
>
> > Q: Could the authors run the experiments for more iterations?
>
> Due to computational resource limitation, we only run 40M frames on 8 atari games. We will add experimental results with more iterations (200M) in our camera ready version.
>
> [1] Foerster, J. N., Chen, R. Y., Al-Shedivat, M., Whiteson, S., Abbeel, P., & Mordatch, I. (2017). Learning with opponent-learning awareness. arXiv preprint arXiv:1709.04326.
>
> [2] Zheng, Z., Oh, J., & Singh, S. (2018). On learning intrinsic rewards for policy gradient methods. Advances in Neural Information Processing Systems, 31.
>
> [3] Fallah, A., Mokhtari, A., & Ozdaglar, A. (2020, June). On the convergence theory of gradient-based model-agnostic meta-learning algorithms. In International Conference on Artificial Intelligence and Statistics (pp. 1082-1092). PMLR.
>
> [4] Ji, K., Yang, J., & Liang, Y. (2020). Multi-step model-agnostic meta-learning: Convergence and improved algorithms.
>
> [5] Fallah, A., Georgiev, K., Mokhtari, A., & Ozdaglar, A. (2020). Provably convergent policy gradient methods for model-agnostic meta-reinforcement learning. arXiv preprint arXiv:2002.05135.

---

### Official Review · Reviewer_UkbE · 2022-07-11

**Rating:** 6
**Confidence:** 2
**Soundness:** 3 good
**Presentation:** 3 good
**Contribution:** 3 good

**Summary:**

Gradient-based meta-learning (GMRL) has been a topic of interest in the NeurIPS community recently. The authors derive formal bounds on the bias and variance of the estimated meta-gradient in commonly-used algorithms for gradient-based meta-reinforcement learning, identifying two sources of meta-gradient bias in particular: "compositional bias" (accumulated across inner loop steps estimated with small batch sizes) and "multi-step Hessian bias", caused by using approximate adaptation in the inner loop. Using these observations, the authors applying existing techniques for variance reduction to new GMRL settings, showing that they can reduce bias and improve GMRL performance in some settings.

**Questions:**

How specific are the conclusions in this paper to gradient-based meta-RL? To what extent are they useful for gradient-based supervised meta-learning?

Ideally, the authors might evaluate horizons that are deeper than only 5 gradient steps. Does LVC continue to provide effective mitigation at greater depth?

Most importantly, the experimental evaluations are relatively limited. It's personally difficult for me to judge the extent of the impact of the theoretical results alone, and I therefore would have found it quite interesting to consider other methods than just LVC. Could the authors justify why LVC is a particularly representative method to use for most of the evaluations?

**Limitations:**

More discussion of the limitations of the theoretical results would be welcome (e.g., how realistic are assumptions 1-3?).

**Strengths And Weaknesses:**

Strengths:
- Analysis of a relevant topic of interest to the NeurIPS community (gradient-based meta-RL)
- Intuitive theoretical result motivating application of existing techniques to new meta-RL settings
- Overall clear presentation

Weakness:
- Unclear what the extent of the impact of the theoretical results are
- Little methodological novelty (which could be okay)
- Experimental evaluations are okay, but the choice of methods to evaluate seems arbitrary and perhaps limited (see final question below)

---

> ### Author Response · Authors · 2022-08-02
> **Response to Reviewer UkbE**
>
> Many thanks for the most helpful review! Based on the review comments, we illustrate how realistic the Assumption 4.1-4.3 are in Appendix D, via which we hope that we have addressed the reviewer’s previous suggestions satisfactorily. We will appreciate it very much if the reviewer can go over these revisions and re-evaluate the paper. The updated PDF of our paper has been uploaded.
>
> > Q: How specific are the conclusions in this paper to gradient-based meta-RL?
>
> We identify two source of bias that existing solutions suffer from in gradient-based meta-RL(GMRL). Compositional bias and Multi-step Hessian bias, we give an upper bound of compositional bias $\mathcal{O}\big(K\alpha^{K}\hat{\sigma}\_{\text{In}}|\tau|^{-0.5}\big)$ with respect to the inner-loop update step $K$, the learning rate $\alpha$, the estimate variance $\hat{\sigma}^{2}\_{\text{In}}$ and the sample size $|\tau|$, and we argue that Multi-step Hessian bias grows exponentially as the inner-loop update step increasing and this bias will affect total meta-gradient bias in a polynomial way $\mathcal{O}\big((K-1)(\hat{\Delta}\_{H})^{K-1}\big)$. We verified our two finding through tabular example, and we came up with two specific solution to solve different GMRL algorithms that are suffered from two biases respectively.
>
> > Q: To what extent are they useful for gradient-based supervised meta-learning?
>
> The lemma and theorem derived in our paper are RL-specific so they cannot directly extended to gradient-based supervised meta-learning. However,conclusion regarding compositional bias basically hold in gradient-based supervised meta-learning because of the shared bilevel optimization framework, especially when the inner-loop conducted SGD (but such problem will be more severe in RL becuase RL tends to have much higher variance compared with supervised learning). The conclusion about multi-step hessian bias don't hold in gradient-based supervised meta-learning for the reason that there is no biased Hessian estimation issue in gradient-based supervised meta-learning. Plus, the theoretical analysis presented in 4.3 are considering the coupling effect of compositional bias and multi-step hessian bias, which result in the final gradient bias and variance. This is RL-specific.
>
> > Q: Does LVC continue to provide effective mitigation at greater depth?
>
> We did thorough ablation studies with LVC in section 5.1, LVC can still provide effective mitigation according to Figure 6. in Appendix G.1.3. We will also conduct experiment with greater depth for our Atari experiment in the camera ready version.
>
> > Q: Could the authors justify why LVC is a particularly representative method to use for most of the evaluations?
>
> Theoretically, LVC offers a low-bias, low-variance estimation and this can help with the trade-off and finally get estimation with relatively low estimation error. Empirically, as the result of ablation studies shows in Appendix G.1.3., LVC is a perfect exising solution which achieve great performance on balancing meta-gradient bias and variance in tabular MDP setting, that's why we use LVC as an mitigation solution.
>
> > Q: More discussion of the limitations of the theoretical results would be welcome (e.g., how realistic are assumptions 1-3?).
>
> Many thanks! We have added it in Appendix D.

---

> > ### Comment · Reviewer_UkbE · 2022-08-10
> > **Response to author response**
> >
> > I appreciate the authors' response, particularly the added clarifications regarding the specificity of the results to RL settings, rather than general gradient-based supervised meta-learning, as well as the discussion of the potentially limited applicability of the assumptions used in Appendix D. However, I still would have liked to evaluate the effectiveness of LVC beyond 5/6 inner loop steps, as well as other mitigation solutions than LVC. I will thus keep my score; I believe this is a pretty good paper, and would be okay with it being accepted, but I would definitely be more convinced if the empirical evaluations were more comprehensive.

---

### Official Review · Reviewer_H2bz · 2022-07-13

**Rating:** 6
**Confidence:** 3
**Soundness:** 1 poor
**Presentation:** 2 fair
**Contribution:** 3 good

**Summary:**

The paper presents a theoretical analysis of two sources of bias in the gradient estimations of meta-reinforcement learning methods that perform a two level optimization of an outer meta-learner and an inner gradient-based RL learner. The authors derive an upper bound to the first source of bias, named compositional bias as well as an analysis of the bias of the estimated Hessians coming from the use of auto-diff systems. Finally, the paper presents an empirical evaluation in tabular MDPs that supports the theoretical findings.

**Questions:**

Given the previous doubts on the compositional, I would ask the authors to:
1) Prove that the compositional bias is actually strictly greater than 0.
2) Comment on my second concern, related to the fact that this in not actually a bias, but a finite-sample error that goes to zero with more samples
3) How does the correlation plotted in Figure 1 relate to the compositional bias identifies. It seems to me again, that the plotted measure is related to MSE instead of the bias, reinforcing my doubts on the analysis performed.

**Limitations:**

The authors have discussed their limitations

**Strengths And Weaknesses:**

=========Update after Rebuttle====
My main concern on the validity of the 2 sources of bias has now been addressed and I agree that the bias terms exists and are strictly higher than 0. For this I will gladly raise my score from 3 to 6.

The paper proposes a general study on gradient estimation meta-reinforcement learning.
While the setting itself is interesting to the community I have some major concerns regarding the theoretical results which are listed as follow.
A) On the gradient expression. (Proposition 3.1)
I have some concerns regarding the correctness of the proof.
In particular, I would ask the authors to derive step by step  Equation 18 as according to my computation, the total derivative has been simplified with the gradient, which, in general is not correct.
B) On the compositional bias. (Section 4.1)
To the best of my understanding, the authors did not show that this term actually exists, instead they provide an upper bound on it (i.e., one might take a general unbiased estimator and upper bounds its bias with any positive number).
The second concern regarding the compositional bias is the fact that, as Lemma 4.4 shows, the compositional bias is upper bounded by the empirical variance divided by the number of trajectories taken to estimate the inner loop gradients.
This term tends to zero as the number of trajectories tends to infinity. Similar results holds (in high probability) for standard policy gradient setting (see for instance "Adaptive Batch Size for Safe Policy Gradients" (Papini 2017) or "Smoothing Policies and Safe Policy Gradients" (Papini 2019)): the error (not bias) in the gradient comes from a term that depends on the variance and the number of samples used (this can also be seen as a consequence of the Chebyshev inequality). This seems very similar to what the authors claim. In this sense, I am a bit confused on the value of Lemma 4.4 (which has a direct impact on the main theoretical result). More specifically, it seems to me that rather than a bias we are just considering an error in the gradient estimation (that, of course, occurs with finite samples).

C) On the multi-step hessian bias. (Section 4.2)
It is not clear to me where this bias comes from and seems simply to be a bug of differentation software. The discussion of Section 4.2 is  unclear to me.

D) On the trade-off (Section 4.3)
Generally speaking, Theorem 4.5 seems to be auto-referential.
The contribution of multi-step hessian bias is direct: it just states that this error exists and it is accounted in the computation.
The same holds for the compositional bias.
Regarding the variance, instead, the results seems incredibly loose since it does not tend to 0 (V1 * (K-1)) (more than a worst case solution).
Moreover, generally speaking, Theorem 4.5 shows very little insights and its contribution seems to be minor w.r.t. Lemma 4.4.

---

> ### Author Response · Authors · 2022-08-02
> **Response to Reviewer H2bz (1)**
>
> Many thanks for the most helpful review! Based on the review comments, we have rewritten Lemma 4.4 and Section 4.2, via which we hope that we have addressed the reviewer’s previous concerns satisfactorily. We will appreciate it very much if the reviewer can go over these revisions and re-evaluate the paper. The updated PDF of our paper has been uploaded.
>
> > Q1: Step-by-step derivation of Equation (18)
>
> The result of Equation (18) $\nabla_{\boldsymbol{\phi}} J^{K}(\boldsymbol{\phi})
> = \nabla_{\boldsymbol{\phi}} J^{\text {Out}}(\boldsymbol{\phi}, \boldsymbol{\theta}^{K}) + \nabla_{\boldsymbol{\phi}} \boldsymbol{\theta}^{K} \nabla_{\boldsymbol{\theta}^{K}} J^{\text {Out }}(\boldsymbol{\phi}, \boldsymbol{\theta}^{K})$ is a consequence of applying chain rule into composition of two differentiable functions $J^{\text {Out}}(\phi,·)$ and $\boldsymbol{\theta}^{K}(\phi)$. To be more specific, $K$ step inner-loop update results in $\boldsymbol{\theta}^{K}=
> \boldsymbol{\theta}^{0} + \alpha\sum_{i=0}^{K-1} \nabla_{\boldsymbol{\theta}^{i}} J^{\text{In}}(\boldsymbol{\phi},\boldsymbol{\theta}^{i})$, therefore $\boldsymbol{\theta}^{K}$ can be seen as a function of $\phi$, the equation above can be rewrittened as $\boldsymbol{\theta}^{K}(\phi)=
> \boldsymbol{\theta}^{0} + \alpha\sum_{i=0}^{K-1} \nabla_{\boldsymbol{\theta}^{i}} J^{\text{In}}(\boldsymbol{\phi},\boldsymbol{\theta}^{i})$. As mentioned in Equation (1), $J^{K}(\boldsymbol{\phi}):=J^{\text{Out}}(\boldsymbol{\phi}, \boldsymbol{\theta}^{K})$, thus $J^{K}(\boldsymbol{\phi})$ equals $J^{\text{Out}}(\boldsymbol{\phi}, \boldsymbol{\theta}^{K}(\boldsymbol{\phi}))$, which becomes of function of $\boldsymbol{\phi}$. To decompose the gradient term $\nabla_{\boldsymbol{\phi}}J^{\text{Out}}(\boldsymbol{\phi}, \boldsymbol{\theta}^{K}(\boldsymbol{\phi}))$  we can apply the chain rule onto nested function $J^{\text{Out}}(\boldsymbol{\phi}, \boldsymbol{\theta}^{K}(\boldsymbol{\phi}))$, which gives us summation of two separate terms, $\nabla_{\boldsymbol{\phi}} J^{\text {Out}}(\boldsymbol{\phi}, \boldsymbol{\theta}^{K})$ and $\nabla_{\boldsymbol{\phi}} \boldsymbol{\theta}^{K} \nabla_{\boldsymbol{\theta}^{K}} J^{\text {Out }}(\boldsymbol{\phi}, \boldsymbol{\theta}^{K})$. Second term is the direct result of chain rule over gradient of multivariable composite function, which gives us form of Equation (18).
>
> Similar form can be seen in wide series of paper(i.e. Multi-step MAML[1] Equation after Equation (64), TayPO-2[2] Equation (2), LSF [3] Proof of Lemma 17 Second Equation).
>
> > Q2: The authors did not show that compositional bias actually exists & Prove that the compositional bias is actually strictly greater than 0
>
> We think the main misunderstanding comes from how we define $\hat{\Delta}\_{C}$. We modify the definition from $\hat{\Delta}\_{C} = \mathbb{E} [\| \hat{\boldsymbol{\theta}}^{K}  -  \boldsymbol{\theta}^{K}\| ]$ to $\hat{\Delta}\_{C}= \mathbb{E} [\| f(\hat{\boldsymbol{\theta}}^{K})  -  f(\boldsymbol{\theta}^{K})\| ]$. We also revise the corresponding lemma part and please check our revised paper for more details. The most important and interesting part here is that the compositional bias comes from the adaptation error $\mathbb{E} [\| \hat{\boldsymbol{\theta}}^{K}  -  \boldsymbol{\theta}^{K}\| ]$, which is caused by inner-loop estimation error. In the following we offer a more detailed illustration.
>
> We agree with the fact that if we use infinite samples in the inner-loop optimization under the assumption that function $f(\cdot)$ is Lipschitz continuous, the compositional bias can be reduced to zero. **However**, this is not the point we want to deliver. Compositional bias is a bias issue coming from meta gradient estimation in the scope of **bi-level optimization**. Because of the two stochastic estimation we need to get, things in bi-level optimization are different from that in normal stochastic optimization. The point we want to emphasize is that even if we can get unbiased inner-loop optimization (with finite inner-loop samples) and unbiased outer-loop optimization (with finite outer-loop samples), simply combining these two unbiased estimation in **GMRL** will still result in final biased meta-gradient estimation. This is because the gradient is evaluated at the estimated adapted parameter rather than the expected adapted parameter, while only the latter one can derive the unbiased meta-gradient. For instance, in the following equations, the left side is the expectation of estimation while the right side is the true term in meta-gradient.
> $$
> \mathbb{E}[\nabla_{\hat{\boldsymbol{\theta}}^{K}} J^{\text {Out}}(\boldsymbol{\phi}, \hat{\boldsymbol{\theta}}^{K})] \neq
> 	\nabla_{\boldsymbol{\theta}^{K}} J^{\text {Out }}(\boldsymbol{\phi}, \boldsymbol{\theta}^{K})
> $$
> $$
> \mathbb{E}[\nabla_{\boldsymbol{\phi}} J^{\text {Out}}(\boldsymbol{\phi}, \hat{\boldsymbol{\theta}}^{K})]
> 	\neq\nabla_{\boldsymbol{\phi}} J^{\text {Out }}(\boldsymbol{\phi}, \boldsymbol{\theta}^{K})
> $$

---

> > ### Author Response · Authors · 2022-08-02
> > **Response to Reviewer H2bz (2)**
> >
> > > Q3: C) On the multi-step hessian bias.
> >
> > Multi-step hessian bias is not a bug of differentation software. In MAML-RL, the Hessian bias problem is a classical research problem and has been widely studied in many MAML-RL papers [2,4,6]. Gradient of expectation of the RL-objective are generally computed with a Monte Carlo estimate based on the policy gradient theorem. However, hessian estimate obtained by automatically differentiating a normal policy gradient surrogate objective results in **biased hessian estimate**. This is because RL-objective are defined via expectations with respect to the distribution of policy trajectory. Here we offer a more detailed deriviation about the Hessian problem in one-step MAML-RL. **We also add more background knowledge of this part into the Appendix C for better understanding.**
> >
> > We denote the initial parameter as $\boldsymbol{\theta}^{0}$ and the adapted parameter as $\boldsymbol{\theta}^{1}$. We can get the inner-loop process and outer-loop meta-gradient expectation form:
> > \begin{equation}
> >     \boldsymbol{\theta}^{1} = \boldsymbol{\theta}^{0} + \alpha \mathbb{E}\_{\boldsymbol{\tau} \sim p(\boldsymbol{\tau} ; \boldsymbol{\theta}^{0})}[\nabla_{\boldsymbol{\theta}^{0}} \log \pi(\boldsymbol{\tau} ) \mathcal{R}(\boldsymbol{\tau})]
> > \end{equation}
> > \begin{equation}
> > \nabla_{\boldsymbol{\theta}^{0}} \boldsymbol{\theta}^{1}=I+\mathbb{E}\_{\boldsymbol{\tau} \sim p(\boldsymbol{\tau} ; \boldsymbol{\theta}^{0})}\left[\mathcal{R}(\boldsymbol{\tau})\left(\nabla_{\boldsymbol{\theta}^{0}}^{2} \log \pi_{\boldsymbol{\theta}^{0}}(\boldsymbol{\tau})+\nabla_{\boldsymbol{\theta}^{0}} \log \pi_{\boldsymbol{\theta}^{0}}(\boldsymbol{\tau}) \nabla_{\boldsymbol{\theta}^{0}} \log \pi_{\boldsymbol{\theta}^{0}}(\boldsymbol{\tau})^{\top}\right)\right]
> > \end{equation}
> >
> > Typically we need to use trajectory samples $\boldsymbol{\tau}{n}$ to estimate the policy gradient, we can get the adapted policy estimate.
> > \begin{equation}
> >     \hat{\boldsymbol{\theta}}^{1} = \boldsymbol{\theta}^{0} + \alpha \frac{1}{N} \sum_{\boldsymbol{\tau}{n}} \sum_{t=0}^{H-1} \nabla_{\boldsymbol{\theta}}\log \pi\_{\boldsymbol{\theta}}(\boldsymbol{a}^{n}\_{t} \mid \boldsymbol{s}^{n}\_{t})\left(\sum_{t^{\prime}=0}^{H} \gamma^{t} r\left(\boldsymbol{s}^{n}\_{t^{\prime}}, \boldsymbol{a}^{n}\_{t^{\prime}}\right)\right)
> > \end{equation}
> > Finally, implementation of MAML-RL derives the gradient estimate by automatic differentation. The corresponding estimation is biased:
> > \begin{equation}
> > \begin{aligned}
> > \mathbb{E}[\nabla_{\boldsymbol{\theta}^{0}} \hat{\boldsymbol{\theta}}^{1}]&=I+\alpha\mathbb{E}\_{\boldsymbol{\tau} \sim p(\boldsymbol{\tau} ; \boldsymbol{\theta}^{0})}\left[\frac{1}{N} \sum_{\boldsymbol{\tau}{n}} \sum_{t=0}^{H-1} \nabla_{\boldsymbol{\theta}^{0}}^{2}\log \pi\_{\boldsymbol{\theta}^{0}}(\boldsymbol{a}\_{t}^{n} \mid \boldsymbol{s}^{n}\_{t})\left(\sum_{t^{\prime}=0}^{H} \gamma^{t} r(\boldsymbol{s}^{n}\_{t^{\prime}}, \boldsymbol{a}^{n}\_{t^{\prime}})\right)\right] \\
> > &=I+\alpha\mathbb{E}\_{\boldsymbol{\tau} \sim p(\boldsymbol{\tau} ; \boldsymbol{\theta}^{0})}\left[\mathcal{R}(\boldsymbol{\tau})\nabla_{\boldsymbol{\theta}}^{2} \log \pi_{\boldsymbol{\theta}^{0}}(\boldsymbol{\tau})\right]\not=\nabla_{\boldsymbol{\theta}^{0}}\boldsymbol{\theta}^{1}
> > \end{aligned}
> > \end{equation}
> > Build upon this, in our paper we mainly want to deliver 3 points: (1) Beyond MAML-RL, many gradient-bsed Meta-RL paper are suffering from the same problem but they lack theoretical analaysis and empirical verification. (2) Some meta gradient estimation coincidentally to be unbiased in on-step scenario, which is mentioned in section 4.2 line 203-204: ** We can see from Eq. (7) that the it would not involve Hessian $\nabla^{2}J^{\text {In }}$ computation if $\boldsymbol{\phi}\neq \boldsymbol{\theta}$. The reason why our section is called **Multi-step hessian bias** is mentioned in section 4.2 line 203-204: we can see from Eq. (7) that the it would not involve Hessian $\nabla^{2}J^{\text {In }}$ computation if $\boldsymbol{\phi}\neq \boldsymbol (3) The Compositional bias presented (section 4.1) will have coupling effect over the Hessian bias (section 4.2), which is further discussed in section 4.3.

---

> > > ### Author Response · Authors · 2022-08-02
> > > **Response to Reviewer H2bz (3)**
> > >
> > > > Q4: D) On the trade-off (Section 4.3)
> > >
> > > Theorem 4.5 is not auto-referential. Regarding the upper bound of bias, first multi-step hessian bias has polynomial impact on meta-gradient bias, term $(B + \hat{\Delta}{H})^{K-1}$, $K$ is the inner-loop update step. Also, compositional bias and multi-step hessian bias can have coupling effect on the meta-gradient bias, there are terms that multiplies compositional bias and multi-step hessian bias together, the reason why is that we use $\nabla_{\boldsymbol{\phi}}\hat{\boldsymbol{\theta}}^{K} \nabla_{\hat{\boldsymbol{\theta}}^{K}}
> > > \hat{J}^{\text {Out}}(\boldsymbol{\phi},\hat{\boldsymbol{\theta}}^{K},\boldsymbol{\tau}3)$ to estimate $\nabla_{\boldsymbol{\phi}}\boldsymbol{\theta}^{K} \nabla_{\boldsymbol{\theta}^{K}}
> > > J^{\text {Out}}(\boldsymbol{\phi}, \boldsymbol{\theta}^{K})$, according to the analytical form of $\nabla_{\boldsymbol{\phi}}\boldsymbol{\theta}^{K}$ in equation (3) and $\nabla_{\boldsymbol{\phi}}\hat{\boldsymbol{\theta}}^{K}$ in equation (4), compositional bias happens in estimating $\nabla_{\boldsymbol{\theta}^{K}}
> > > J^{\text {Out}}(\boldsymbol{\phi}, \boldsymbol{\theta}^{K})$ and multi-step hessian bias happens in estimating $\nabla_{\boldsymbol{\phi}}\boldsymbol{\theta}^{K}$, estimating the above two terms multiplying together results in coupling effect on the meta-gradient bias. Regarding the upper bound of variance, the reason why terms like $(K-1)$ exists is another evidence of why we name it **multi-step hessian bias**. There is no hessian bias in meta-gradient bias when $K=1$, the above statement also holds in upper bound of bias.
> > >
> > > > Q5: How does the correlation plotted in Figure 1 relate to the compositional bias identifies.
> > >
> > > Our initial motivation for experiment 1 is to illustrate how different gradient estimation conditions can affect the final meta-gradient estimation. That is why we finally pick correlation for presenting the ablation study. And we also offer the meta-gradient varaince plot in figure 4(Appendix G.1.3), which can further help us with the meta-gradient variance analysis of influence brought by compositional bias and Hessian bias. We are also working on updating the experimental results to further derive the bias plot in different estimation conditions for clearer presentation and stronger connection with our theory.
> > >
> > >
> > > [1] Ji, K., Yang, J., & Liang, Y. (2020). Multi-step model-agnostic meta-learning: Convergence and improved algorithms.
> > >
> > > [2] Tang, Y., Kozuno, T., Rowland, M., Munos, R., & Valko, M. (2021). Unifying gradient estimators for meta-reinforcement learning via off-policy evaluation. Advances in Neural Information Processing Systems, 34, 5303-5315.
> > >
> > > [3] Tang, Y. (2022, June). Biased gradient estimate with drastic variance reduction for meta reinforcement learning. In International Conference on Machine Learning (pp. 21050-21075). PMLR.
> > >
> > > [4] Rothfuss, J., Lee, D., Clavera, I., Asfour, T., & Abbeel, P. (2018). Promp: Proximal meta-policy search. arXiv preprint arXiv:1810.06784.
> > >
> > > [5] Fallah, A., Georgiev, K., Mokhtari, A., & Ozdaglar, A. (2021). On the convergence theory of debiased model-agnostic meta-reinforcement learning. Advances in Neural Information Processing Systems, 34, 3096-3107.
> > >
> > > [6] Liu, H., Socher, R., & Xiong, C. (2019, May). Taming maml: Efficient unbiased meta-reinforcement learning. In International conference on machine learning (pp. 4061-4071). PMLR.

---

> ### Comment · Reviewer_H2bz · 2022-08-07
> **Response to rebuttle**
>
> I thank the authors for the thorough response and apologize for the delay on my response as I needed some days to digest the responses.
>
> On Q2 and Q3, I thank the authors for the detailed responses and the details added to the paper to clarify them and make them more readable and understandable. My main concern on the validity of the 2 sources of bias has now been addressed and I agree that the bias terms exists and are strictly higher than 0.  For this I will gladly raise my score from 3 to 5.
>
> There are still some open concerns that remain for me.
>
> 1. I still have some questions regarding Q1. I agree with the derivations in the authors response. What Is not clear to me at the moment is the application of the last chain rule. I would not define myself as an expert in multivariate calculus, but in my understanding the application of the last chain rule would also include a term related to the total derivative as in Equation [3] in [1].
> 2. The relation of the theory to the experiments. I would welcome the addition of the plots actually estimating each source of bias for a better link between the theory and the experiments of the paper. Plotting MSE instead of the bias terms only adds to the problems of readability of the paper.
>
> [1] Zhongwen Xu, Hado van Hasselt, David Silver, Meta-Gradient Reinforcement Learning. NeurIPS 2018: 2402-2413

---

> > ### Author Response · Authors · 2022-08-08
> > **Response to Reviewer H2bz (4)**
> >
> > Really thank you for your insightful suggestions, here are our responses:
> >
> > > Q: What Is not clear to me at the moment is the application of the last chain rule. but in my understanding the application of the last chain rule would also include a term related to the total derivative as in Equation [3] in [1].
> >
> > Equation [3] is actually a supporting evidence of **Equation (22)** (Equation (18) before the revision) and **Equation (23)**  in our paper. To be more specific, Equation [3] takes the same form of Equation (23) without the gradient direction in **one-step** scenario. ($K=1$).
> >
> > When $K=1$, Equation (22) can be written as:
> >
> > \begin{equation}
> > \begin{aligned}
> >     \nabla_{\boldsymbol{\phi}} J^{1}(\boldsymbol{\phi})
> >     &= \nabla_{\boldsymbol{\phi}} J^{\text {Out}}(\boldsymbol{\phi}, \boldsymbol{\theta}^{1}) + \nabla_{\boldsymbol{\phi}} \boldsymbol{\theta}^{1} \nabla_{\boldsymbol{\theta}^{1}} J^{\text {Out}}(\boldsymbol{\phi}, \boldsymbol{\theta}^{K})\\
> > \end{aligned}
> > \end{equation}
> >
> > We then expand the term $\nabla_{\boldsymbol{\phi}} \boldsymbol{\theta}^{1}$. Let $i=0$ in Equation (23), we can get:
> >
> > \begin{equation}
> > \nabla_{\boldsymbol{\phi}} \boldsymbol{\theta}^{1}
> > =\nabla_{\boldsymbol{\phi}} \boldsymbol{\theta}^{0} \left(I+\alpha \nabla_{\boldsymbol{\theta}^{0}}\nabla_{\boldsymbol{\theta}^{0}}J^{\text{In}}(\boldsymbol{\phi}, \boldsymbol{\theta}^{0})\right) + \alpha \nabla_{\boldsymbol{\phi}} \nabla_{\boldsymbol{\theta}^{0}} J^{\text{In}}(\boldsymbol{\phi}, \boldsymbol{\theta}^{0})\\
> > \end{equation}
> >
> > The above equation still holds when we drop the gradient direction, we can then replace the gradient symbol $\nabla$ with partial derivative symbol $\partial$:
> >
> > \begin{equation}
> >  \frac{\partial \boldsymbol{\theta}^{1}}{\partial {\boldsymbol{\phi}}}
> > = \frac{\partial \boldsymbol{\theta}^{0}}{\partial {\boldsymbol{\phi}}} \left(I+\alpha \frac{\partial^{2} J^{\text{In}}(\boldsymbol{\phi}, \boldsymbol{\theta}^{0})}{\partial \boldsymbol{\theta}^{0} \partial \boldsymbol{\theta}^{0}}\right) + \alpha \frac{\partial^{2} J^{\text{In}}(\boldsymbol{\phi}, \boldsymbol{\theta}^{0})}{\partial \boldsymbol{\phi} \partial \boldsymbol{\theta}^{0}}\\
> > \end{equation}
> >
> > Rearrange the equation, we can get:
> >
> > \begin{equation}
> >  \frac{\partial \boldsymbol{\theta}^{1}}{\partial {\boldsymbol{\phi}}}
> > =  \left(I+\alpha \frac{\partial^{2} J^{\text{In}}(\boldsymbol{\phi}, \boldsymbol{\theta}^{0})}{\partial \boldsymbol{\theta}^{0} \partial \boldsymbol{\theta}^{0}}\right)\frac{\partial \boldsymbol{\theta}^{0}}{\partial {\boldsymbol{\phi}}} + \alpha \frac{\partial^{2} J^{\text{In}}(\boldsymbol{\phi}, \boldsymbol{\theta}^{0})}{\partial \boldsymbol{\phi} \partial \boldsymbol{\theta}^{0}}\\
> > \end{equation}
> >
> > According to the Equation (1) in [1] $\theta^{\prime}=\theta+f(\tau, \theta, \eta)$ and inner-loop update dropping gradient direction $\boldsymbol{\theta}^{1}=
> > \boldsymbol{\theta}^{0} + \alpha \frac{\partial J^{\text{In}}(\boldsymbol{\phi}, \boldsymbol{\theta}^{0})}{\partial \boldsymbol{\theta}^{0}}$, then we change $\boldsymbol{\theta}^{1}$ in above equation with $\theta^{\prime}$, $\boldsymbol{\theta}^{0}$ with $\theta$, $\boldsymbol{\phi}$ with $\eta$. Combined with the fact that $J^{\text{In}}(\boldsymbol{\phi}, \boldsymbol{\theta}^{0})=\mathbb{E}\_{\boldsymbol{\tau} \sim p(\boldsymbol{\tau} ; \boldsymbol{\theta}^{0})}
> >     \left[
> > \sum_{t=0}^{H-1}
> >     \nabla_{\boldsymbol{\theta}^{0}}\log\pi_{\boldsymbol{\theta}^{0}}(\boldsymbol{a}\_{t}|\boldsymbol{s}\_{t})
> > \left(g_{\boldsymbol{\phi}}(\boldsymbol{\tau})-v_{\boldsymbol{\theta}^{0}}(\boldsymbol{s}_{t})\right)\right]$ in MGRL, we can change $\alpha \frac{\partial J^{\text{In}}(\boldsymbol{\phi}, \boldsymbol{\theta}^{0})}{\partial \boldsymbol{\theta}^{0}}$ with $\partial f(\tau, \theta, \eta)$, then we can see the form of Equation [3] in [1] reveals:
> >
> > $$
> > \frac{\mathrm{d} \theta^{\prime}}{\mathrm{d} \eta}=\left(I+\frac{\partial f(\tau, \theta, \eta)}{\partial \theta}\right) \frac{\mathrm{d} \theta}{\mathrm{d} \eta}+\frac{\partial f(\tau, \theta, \eta)}{\partial \eta}
> > $$
> >
> > [1] Zhongwen Xu, Hado van Hasselt, David Silver, Meta-Gradient Reinforcement Learning. NeurIPS 2018: 2402-2413

---

> > > ### Author Response · Authors · 2022-08-08
> > > **Response to Reviewer H2bz (5)**
> > >
> > > > Q: Plotting MSE instead of the bias terms only adds to the problems of readability of the paper.
> > >
> > > Thank you for pointing our the problem. In these days we add more experiments over ths bias term to make the clearer presentation and stronger connection with our theory. We update our result in **Appendix J.1.3 (Figure 5)** and here is a summarization of our additional experiments: In the first experiment, we successfully validate our Lemma 4.4 ($\mathcal{O}\big(K\alpha^{K}\hat{\sigma}\_{\text{In}}|\tau|^{-0.5}\big)$) about the **exponential** impact brought by the inner-loop step $K$ and in the second experiment,  we conduct experiments to verify the **polynomial** impact $\mathcal{O}\big((K-1)(\hat{\Delta}\_{H})^{K-1}\big)$ on the meta-gradient bias introduced by the multi-step Hessian estimation bias $\hat{\Delta}\_{H}$.
> > >
> > > We are also working on further experiments considering ablation study over different estiamtion conditions(learning rate/estimator/sample size) and will update the paper as soon as possible.
> > >
> > > In addtion, we also believe that our experiments in Section 5.2, Section 5.3 are also our important contributions on real world RL problem. It takes especially a long time for us to reproduce MGRL presented in Section 5.3 since it is not open-source. We hope the reviewer can also take these experiments into consideration. We hope our response can help you further understand our paper. Thank you!

---

> > > > ### Comment · Reviewer_H2bz · 2022-08-09
> > > > **Rensponse to author**
> > > >
> > > > I thank the reviewers for the additional clarification on Equation 18 and the additional experiments.
> > > > I think the additional experiments directly linking the theory to the experiments (validating the theory empirically) greatly increase the quality and readability of the paper. For these reason I will gladly increase the score to 6.

---

### Author Response · Authors · 2022-08-07
**Looking forward to further discussion**

Dear Reviewers,
We greatly appreciate your insightful comments on our paper. We respectfully look forward to following up to see if the authors' response has addressed your concerns or if you have any further questions. Thank you!

---

### Meta-Review · Area_Chair_U974 · 2022-08-29

**Recommendation:** Accept
**Confidence:** Certain

**Metareview:**

All reviewers are positive about this paper and recommend either weak (3x) or borderline (1x) accept. We had some good discussions, through which some of the reviewers (RjbJ and H2bz) decided to increase their initial scores. The reviewers consider this work a relevant topic, providing intuitive and novel theoretical results, well-written, and with the possibility of guiding future algorithmic design for meta-RL. I believe this is a good paper, and I recommend its acceptance.

**Award:**

No

---

### Decision · Program_Chairs · 2022-09-14

Accept